

# Technical assessment combined with extended cost-benefit analysis for groundwater ecosystem services restoration - An application for Grand Bahama

Anne Imig[1], Francesca Perosa[2], Carolina Iwane Hotta[2], Sophia Klausner[1], Kristen Welsh[3,4], Arno Rein[1*]

[1]Chair of Hydrogeology, School of Engineering and Design, Technical University of Munich, Germany; anne.imig@tum.de, sophia.klausner@tum.de, arno.rein@tum.de
[2]Chair of Hydrology and River Basin Management, School of Engineering and Design, Technical University of Munich, Germany; francesca.perosa@tum.de, Carolina.hotta@tum.de
[3]Small Island Sustainability Programme, University of The Bahamas, Nassau, Bahamas; kristen.unwala@ub.edu.bs

[4]Geosciences Department, Oberlin College, Oberlin, Ohio, USA; k.welsh@oberlin.edu

*Correspondence to*: Arno Rein arno.rein@tum.de

**Abstract.** A large storm surge caused by Hurricane Dorian in 2019 resulted in extensive flooding and saltwater intrusion into
the aquifers of Grand Bahama Island. This caused 40% of the island's water supply to become brackish with no or slow recovery to date and damage of more than 70 % of mangroves and forests on Grand Bahama. Managed aquifer recharge (MAR) and reforestation were considered as nature-based solutions to mitigate the impacts of Hurricane Dorian. First, a technical assessment of MAR investigated (hydro-)geological aspects. As a result, potential locations for a MAR scheme are proposed. Further, a financial and an extended cost-benefit analysis (CBA) integrating ecosystem services (ES) assessments are
conducted for proposed MAR and reforestation measures. Based on the current data availability, results indicate that the MAR scheme of rooftop rainwater harvesting is technically feasible. However, based on our first estimate with limited data, this measure will be able to provide only about 10% of water demand in the study area and thus would not be favorable from a financial perspective. Since MAR has a range of positive aspects (including potential reduction of desalinization efforts and improvement freshwater-dependent ecosystems), we recommend reassessment with more detailed hydrogeological data. On
the other hand, reforestation measures are assessed as financially profitable. The results of this study prove the technical feasibility and the added value of restoring the groundwater ecosystem on Grand Bahama, but also highlight the associated high costs.



## 1. Introduction

The consequences of the Anthropocene, in particular climate change and resulting impacts, are negatively affecting small islands and their water resources. These effects will continue to be observed in the future decades (Thomas et al., 2020). Freshwater aquifers on small islands manifest as thin lenses and are sustained solely by recharge from rainfall. The freshwater lenses float atop more saline groundwater from seawater (Ault, 2016; Bedekar et al., 2019). Wave-induced overwash leads to the infiltration of saltwater into the freshwater lenses, which becomes more frequent with sea-level rise and increasing

frequency and intensity of hurricanes (Emanuel, 2020; Terry and Falkland, 2010; Vecchi et al., 2021). Both island inhabitants and (forest) ecosystems rely heavily on these fragile and limited aquifers, making this resource finite and vulnerable (Diamond and Melesse, 2016; Morgan and Werner, 2014).

Grand Bahama (GB) is a primarily low-lying island in the archipelago of The Bahamas and in the North Atlantic hurricane belt. The island is particularly vulnerable to sea level rise and wave-induced overwash events because approximately 80% of

the land surface elevation is lower than 1 m a.s.l. (Department of Statistics, 2012; ICF and BEST, 2001; Whitaker and Smart, 1997).

Hurricane Dorian struck GB in September 2019. It was one of the most devastating natural disasters that The Bahamas has experienced to date, with damage worth a quarter of the country's gross domestic product. It stalled over GB and the neighboring island of Abaco for more than 24 hours, exerting extremely high wind speeds and covering more than half of GB

under its storm surge and associated flooding (SWA and WES, 2019; UNECLAC, 2021; Zegarra et al., 2020). This resulted in widespread saltwater intrusion into the shallow freshwater aquifers. Approximately 40% of the water supply became brackish, and 30% of the population still lacks a supply with potable water, to date. In addition, stagnant saline water and high wind caused extensive destruction of trees, mangroves, seagrass beds, and coral reefs. It was estimated that 73% of the mangrove habitats and 77% of the forests in GB were damaged (Bahamas National Trust, 2020). Al Baghdadi (2021) predicted

the natural recovery of the groundwater system to take approximately 20 years.

Efforts were initiated to mitigate the devastating impacts of Hurricane Dorian on groundwater and the forest ecosystem, as these provide services of immense societal and economic value. Freshwater aquifers are the only source of drinking water supply on GB to sustain the water demand of the local population and the economy, primarily based on tourism. Further, forest protection and restoration are critical for mitigating climate change and its impacts (van Oosterzee et al., 2020) and stabilizing

groundwater recharge and quality (Ellison, 2018).

After Hurricane Dorian, the Grand Bahama Utility Company (GBUC) announced the installation of a reverse osmosis (RO) system to reduce water salinity to World Health Organization (WHO) standards and create a sustainable and resilient contingency plan in the event of another hurricane (GBUC 2021, 2020). The RO system was fully operating from December 2021, but up to date [October 2023], the water supplied to some households is not yet potable according to WHO standards.

Apart from the shortcomings in the quality of supplied water for drinking water purpose, pipes and faucets are corroding in the Bahamians' households due to the water's high salinity. Further, RO is a highly energy-consuming technology for drinking



water treatment. Consequently, a major concern for the system is the cost dictated by energy consumption, added to the membrane replacement costs (Dillon, 2005; Garfí et al., 2016).

Furthermore, hurricanes can severely damage infrastructure and cause disruptions in the energy supply, leading to damage or

inoperability of the RO system. Therefore, alternative, complementary measures should be used to restore and preserve the existing freshwater resources instead of entirely depending on desalination.

Nature-based solutions (NBS) could be an approach to maintain the drinking water supply on GB and restore the forest ecosystem sustainably and resiliently. According to Cohen-Shacham et al. (2016), NBS are actions to manage and restore natural ecosystems that address societal challenges (e.g., climate change, water security, or natural disasters) effectively and

adaptively, providing human well-being and biodiversity benefits. NBS are gaining acceptance as a more sustainable solution to mitigate and adapt to the effects of climate change by reducing exposure to natural hazards and vulnerability to hazardous events (Sudmeier-Rieux et al., 2021, 2019). They are considered cost-effective and viable solutions to optimize the properties of natural ecosystems and can be integrated with technological and engineering solutions (Cohen-Shacham et al., 2016; Lupp et al., 2021).

Within this study, two planned NBS measures on GB are assessed to mitigate the impacts of Hurricane Dorian on the groundwater ecosystem in GB: managed aquifer recharge and reforestation. Managed aquifer recharge (MAR) is a NBS increasingly deployed in the last decades to tackle saltwater intrusion and climate change effects on groundwater resources. Excess water from other sources, e.g., rainfall/flooding, water treatment plants, rivers, or desalinated seawater, can infiltrate an aquifer to store and recharge groundwater (e.g., Dillon et al., 2019; Gale, 2005; Raicy et al., 2012). For small islands, reports

on MAR implementation are scarce (Hejazian et al., 2017a).

CBA analysis has been applied in existing literature to assess the economic feasibility of MAR projects (e.g., Halytsia et al., 2022; Rupérez-Moreno et al., 2017) but has not included ecosystem services: one of the highlighted benefits of NBS. Furthermore, the CBA method falls short to adequately monetarize ecosystems services (e.g.,Maliva, 2014; Ruangpan et al., 2020; Network Nature, 2022; Sudmeier-Rieux et al., 2021; Wegner and Pascual, 2011). We therefore propose a methodology

in this study which sets itself apart from already published research as it aims to combine a technical feasibility assessment and use the results to assess them in an extended cost-benefit analysis (CBA) with ecosystem services analysis. Ecosystem services are modelled with the InVEST software (Sharp et al., 2020).

In this work, first, the technical feasibility of a MAR scheme is investigated. Then the feasibility of MAR and reforestation measures is explored and compared to the RO scheme from an economic and ecosystem services perspective. Additionally,

the ability of a MAR scheme to replace RO-based water supply is explored. This study aims to show methods for investigating ecosystem services from an economic perspective. Results aim to show financial benefits of NBS to policy and decision makers and help justify their implementation.



## 2. Materials and Methods

### 2.1 Study Site

Grand Bahama is the northernmost island of The Bahamas. The Bahamian archipelago, with approximately 700 shallow islands and 2400 cays, is scattered over a strip of approximately 1000 km from the coast of Southern Florida in the North down to Cuba and Haiti in the South (Figure 1). All islands of The Bahamas consist predominantly of limestone, leading to long and narrow shapes and low-lying lands (ICF and BEST, 2001; Whitaker and Smart, 1997). This includes GB, an east-west striking elongated island with a maximum elevation of 20.7 m a.s.l.

GB topography represents a gently undulating plain. The southern coast consists mainly of sand beaches with shallow reefs in front of a deep-sea basin, while the carbonate platform extends further into the northern coast, creating mangrove marshlands. The climate is classified as marine tropical, with dry winters, wet summers, and a hurricane season from June to November (USACE, 2004; Whitaker and Smart, 1997). GB vegetation is typical for the northern Bahamian islands. It consists of Caribbean Pine forests and Palmetto Palm trees in the inland, broad-leaf coppice with hardwood species (especially at the

windward coasts), and mangrove swamps along protected, shallow coasts. Since World War II, the primary industry on GB has been tourism, followed far by banking, fishing, and agriculture (ICF and BEST, 2001). The last census in 2010 revealed a total population of around 350,000 in The Bahamas, of which 51,000 (14,5 %) inhabitants live in GB, with a rising trend compared to preceding years (Department of Statistics, 2012).

GB's potable water supply is entirely supplied by groundwater. Surface water is not available on the island. The average

abstraction rate is estimated to be 26,497 m$^3$/d (7 million gallons per day [mgd]), with approximately 11,356 m$^3$/d (3 mgd) from Wellfield 6 and 15,141 m$^3$/d (4 mgd) from Wellfields 1, 3 and 4 (Figure 1) (personal communication with GBUC). Wellfield 6 is in a low-lying rural area in the southwest of the island and was nearly fully inundated during the storm surge of Hurricane Dorian. Wellfields 1, 3 and 4 are in the city of Freeport in populated areas (Figure 1). All water supply is disinfected with chlorine.

The water from Wellfield 6 (Figure 1) is brackish since Hurricane Dorian's storm surge. For this reason, the water has been treated with a portable reverse osmosis (RO) unit of 3 mgd capacity (equal to 30% of water demand on the island) since the end of 2021. RO is a water treatment option for desalinization in which a partially permeable membrane separates dissolved components in water. The feed water is pressed through the membrane, removing larger dissolved components (UNEP, 1997). The RO scheme is also designed to be mobile as a storm contingency plan (GBUC, 2020).



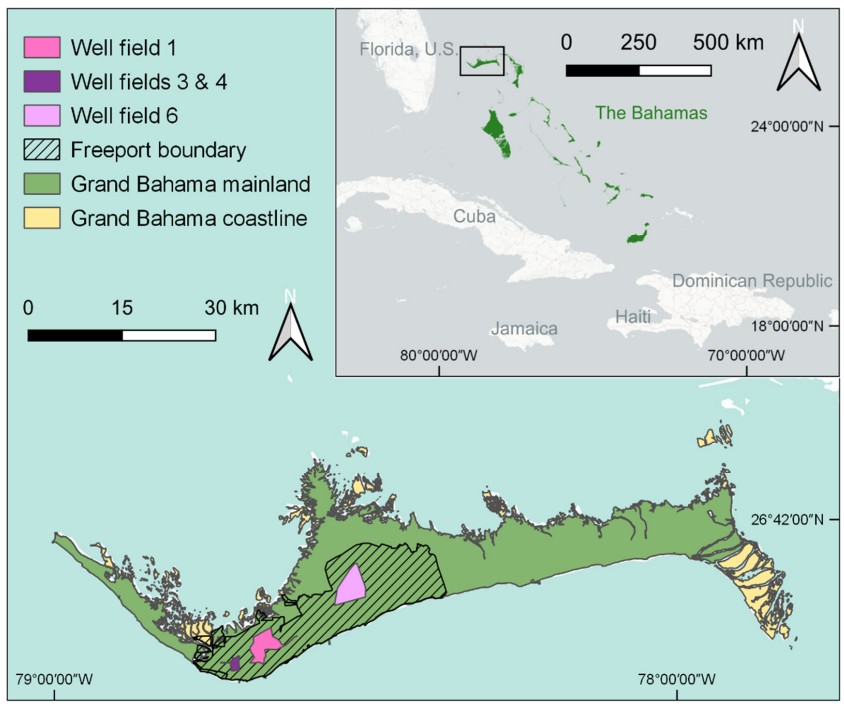

**Figure 1: Location of the study area and wellfields 1, 3, 4, and 6 on the island Grand Bahama, The Bahamas. Freeport boundary (black striped) as well as Grand Bahama mainland (green) and coastline (yellow) are indicated (geographic coordinates: EPSG: 4326).**

## 2.2 Structure of the holistic analysis

Our analysis of potential sustainability measures for GB is based on three main parts (Figure 2). The first part addresses the technical feasibility of potential MAR measures. As an output it provides the information about whether the tested MAR measure is technically feasible, and, if so, which MAR type is the most appropriate. The second part regards the assessment of the financial profitability of the most appropriate MAR measure compared to a portable RO scheme. The latter and other sustainability measures (e.g., reforestation) are assessed by means of a financial cost-benefit analysis (CBA). The third part analyzes the same measures as in the second part, but by means of an extended CBA, i.e., by including ecosystem services as additional benefits.





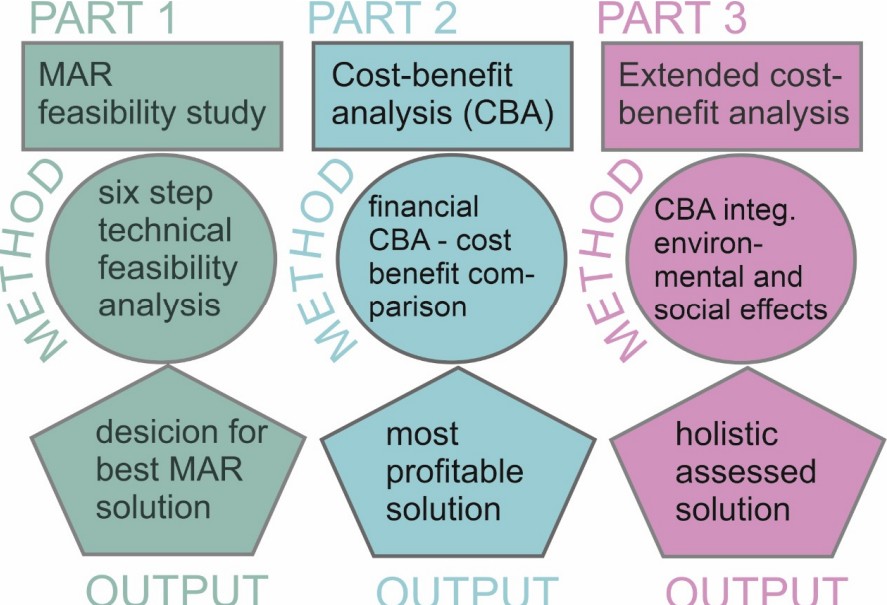

**Figure 2. Flowchart showing the three parts of the holistic assessment for analyzing potential sustainability measures (and comparison to currently applied reverse osmosis) on Grand Bahama.**

**2.3 Sustainability measures**

**Managed aquifer recharge** (MAR) is a nature-based solution (NBS) with the aim for quantitative and qualitative protection of groundwater resources. Excess water from, e.g., rainfall, flooding, water treatment plants, rivers, and desalinated water can be infiltrated into an aquifer to store and recharge groundwater (e.g., Dillon et al., 2019; Gale, 2005; Raicy et al., 2012). As a result, groundwater availability is enhanced, and groundwater can be extracted in a time of need. This measure was analyzed with the technical feasibility procedure, financial CBA, and extended CBA.

**Reforestation** is a NBS measure that implies returning tree cover to deforested land, often intending to reinstate ecological processes at the level of climax forests (Elliott et al., 2013). Moreover, forests are essential in reducing $CO_2$ emissions by carbon sequestration (Piyathilake et al., 2022). The reforestation measure aims at restoring the Bahamian pine trees, already included in an existing seedling nursery on the island (Bowen-O'Connor and Lynch, 2022). This measure was analyzed with the financial CBA and the extended CBA.



### 2.4 Part 1: Technical feasibility assessment of managed aquifer recharge

Various guidelines exist to assess the feasibility of MAR measures in specific study areas. In this study, three independent
guidelines were combined into a new procedure to assess the technical feasibility of MAR on GB. These include a practical
guideline for MAR in the Caribbean developed by a consortium to promote rainwater harvesting (CEHI et al., 2010), the
Australian guidelines for water recycling (NRMMC, 2009, 2006), and a methodology for a feasibility assessment of MAR in
Central Europe (DEEPWATER-CE, 2020a, 2020b). The methodology complies with local requirements such as low-lying
lands exposed to increased hurricane intensity, rising sea levels, and scarce data availability. The methodology is divided into
six main steps that were brought in the form of a decision tree (Figure 3): (i) determination of water demand, (ii) analysis of
suitable aquifers for storage and recovery, (iii) identification of water sources for recharge, (iv) selection of a suitable MAR
type, (v) risk assessment related to the chosen MAR type, and as a final step, (vi) the selection of the most suitable location
for the MAR scheme. If the steps (i)-(iii) and (v) generate a negative evaluation, we suggest extending the study area or
stopping the investigation. Otherwise, if all steps can be followed and result in a positive evaluation, MAR is considered to be
feasible for the study site. Input data used to conduct the technical feasibility assessment (and the other parts of the holistic
analysis) are described in the following sections and in Supplementary Material Table S1.

The water demand can either be defined based on technical guidelines from the country's legislation or based on the
documented water use of the consumers. It is reasonable to predict a water demand for the design life of the MAR measure,
e.g., over 30 years. For the identification of suitable aquifers, the lithology and the location of the aquifer should be studied.
Furthermore, properties such as sufficient storage capacity and hydraulic conductivity should be considered (DEEPWATER-
CE, 2020a; NRMMC, 2006). After defining the water demand and a suitable aquifer, the source(s) for groundwater recharge
should be identified, e.g., rainwater, surface water, or desalinated water. Based on the available water source, a suitable MAR
scheme can be selected for the water demand and the aquifer. This is necessary as, e.g., rainwater harvesting schemes have
different requirements regarding groundwater levels compared to a riverbank filtration scheme (Sallwey et al., 2019). The first
step is to quantitatively determine the water demand and to identify where (geographically) the water is needed. Second,
suitable aquifers for MAR are identified based on chosen criteria (see results section). Consecutively a water source is
identified for the recharge of the MAR scheme. For step (v), we conducted a qualitative risk assessment with a risk score
matrix after Swierc et al. (2005). For step (vi), we developed selection criteria based on information gained from the previous
steps. The criteria were assessed using the geographical information system QGIS (2020), and were used in a multi-criteria
decision analysis (MCDA) (Sallwey et al., 2019). The achievable recharge volume from the rainwater harvesting scheme was
calculated based on recommendations by the German institute for norms (DIN, 2002), where details are given in Section S1
in the Supporting Information (SI).





### 2.4 Part 2: Financial cost-benefit analysis (CBA)

A CBA is a decisional procedure that compares the costs and benefits of a project in monetary terms and uses these quantities
to evaluate the project's effects on the well-being of people (Campos et al., 2018; Clinch, 2004; Hanley, 2013). First, the CBA
approach identifies all costs and benefits of a project; second, it analyses them and assigns monetary values; third, the costs
and benefits are discounted over the lifetime of the project; lastly, the CBA compares costs and benefits with each other
(Hanley, 2013; Nautiyal and Goel, 2021). In this study, the procedures followed to perform the CBA are based on the guidelines
given by the European Commission (2015) for CBA of investment projects. In a CBA, the net present value (NPV) is used to
compare discounted costs and benefits:

$$NPV = present\ value\ of\ benefits – present\ value\ of\ costs \tag{1}$$

A positive NPV indicates that the tested project, measure, or scenario is profitable; otherwise, "the CBA test" failed (Hanley,
2013). Equation (1) can be expressed as follows (Hanley, 2013):

$$NPV \ = \ \sum B_t \frac{1}{(1+r)^t} \ - \ \sum C_t \frac{1}{(1+r)^t} \tag{2}$$


where NPV is determined as the sum of yearly contributions (t = year 1, 2, …, N, where N is the project's lifetime in years)
and B and C respectively represent the benefits and costs of a project. Finally, equation (2) can be rewritten for our purposes
as follows, to represent the results of the financial CBA:

$$NPV_{fin} \ = \ \sum_{t=1}^{N} \frac{1}{(1+r)^t} (DWS - C) \tag{3}$$

DWS represents the benefits of the drinking water supply, C represents the costs, and *r* is the discount rate. In this work, the
project's lifetime was set to 30 years (European Commission, 2015).

The costs of the analyzed sustainability measures were estimated using the analogy method and expert opinion method (Angelis
and Stamelos, 2000). The RO investment costs were based on the published costs (GBUC Public Relations, 2021). Project
manager costs (Supplementary Material Table S1) were based on expert-based knowledge given by the company Phoenix
Engineer (M. Gomez, personal communication, April 14, 2022), while the required hours for each task for the MAR and the
reforestation measures were based on Soža and Patekar (2022).



### 2.5 Part 3: Extended cost-benefit analysis (CBA)

An extended (or social or environmental) CBA includes environmental and other economically relevant impacts in the analysis of a project, implying the valuation of goods and services not exchanged in markets; this is done by using non-market valuation methods (Brouwer and Sheremet, 2017; Clinch, 2004; Hanley, 2013; Martínez-Paz et al., 2014). This approach is more appropriate for evaluating government interventions than a financial CBA. Extended CBAs have already been applied in the past (Acuña et al., 2013; Cerulus, 2014; Grossmann, 2012; Logar et al., 2019; Ruangpan et al., 2020), but application examples are still lacking in the field of MAR. In this work, we present an extended CBA that includes five ecosystem service (ES) types to evaluate the introduced sustainability measures (cf. section 2.2) in a holistic way: (i) drinking water supply and (ii) tourism were included in the extended CBA of RO and MAR scenarios, while (iii) carbon sequestration, (iv) habitat provisioning, and (v) timber provisioning were included in the extended CBA of the reforestation measure. The ES of tourism is a cultural ecosystem service that includes both benefits to visitors and income opportunities for nature tourism service providers (FAO, 2023), as also recognized by the Millennium Ecosystem Assessment (MEA, 2005). The InVEST (Sharp et al., 2020) models "Carbon storage and sequestration" and "Managed timber production" were applied to estimate the biophysical and monetary values of carbon sequestration and timber provisioning, respectively.

The "Carbon storage and sequestration" model is based on the Tier 1 method of the IPCC reports (IPCC, 2014, 2006) . Biophysical carbon sequestration in plant roots and respective carbon storage in a specific region is estimated by aggregating carbon pool values assigned for each land use / land cover (LULC) type (Sharp et al., 2020). For Grand Bahama, the land cover map was reclassified based on the ecofloristic zones defined by the Food and Agriculture Organization (Ruesch and Gibbs, 2008) to differentiate the carbon pools for each zone, leading to 18 carbon classes. The value of carbon sequestration was estimated by multiplying the social cost of carbon (SCC) by the total sequestered carbon. Three different carbon prices were used to address uncertainties in the SCC (Supplementary Material Table S1).

The "Managed timber production" model requires harvest information, including harvest frequency, harvested biomass, and market value of harvested products. As no field data were available for Grand Bahama, the input data were based on previously published literature (Supplementary Material Table S1). As the harvesting frequency of pine trees is usually 30 years, only one harvesting revenue was considered.

It was assumed that the implementation of the sustainability measure, i.e. MAR, would provide potable water for about 30% of the population with a connection to the public water supply (4127 of 13,755 houses connected to public piping), equal to the access to potable water after Hurricane Dorian (Department of Statistics of The Bahamas, 2012).

To estimate the ES of habitat provisioning, we used the willingness-to-pay (WTP) benefit-transfer method to conserve habitat quality, obtained from Wang et al. (2021) (Supplementary Material Table S1). The revenue was calculated by multiplying the WTP and the total number of households in Grand Bahama (Department of Statistics of The Bahamas, 2012). To estimate the value of the tourism ES, we assume that restoring the drinking water supply could increase tourism on Grand Bahama. In fact, tourism facilities (e.g., hotels, restaurants) were also affected by the lack of water supply after Hurricane Dorian, not allowing





them to conduct business in full capacity. In the following we take into account that tourism expenditure would return to the same status as before a hurricane event. Moreover, the tourism sector is affected by a whole range of impacts, where it is complicated to attribute the contribution of the analyzed measures. Accordingly, we estimated the tourism ES of a sustainable measure as 1% of tourism additional revenue (Soža & Patekar, 2022), based on data provided by the Bahamian Tourism Ministry (2022). Therefore, the ES of tourism $T$ can be given as:

$$T = (Average\ tourism\ expenditure\ of\ years\ before\ hurricane\ events \qquad (4)$$
$$- Estimated\ expenditure\ of\ 2021) * 0.01$$

The description of the method applied to estimate annual average tourism expenditure data can be found in Section S2 in the SI. Finally, the NPV of the extended CBA, covering all considered ecosystem services, is estimated by the following equation (modification of Equation 2):

$$NPV_{ext} = \sum_{t=1}^{N} \frac{1}{(1+r)^t} (DWS + Carbon + TP + HP + T - C) \qquad (5)$$

where *Carbon* is the ES of carbon sequestration, *TP* is the ES of timber provisioning, *HP* is the ES of habitat provisioning,
and *T* is the ES of tourism (other definitions cf. Eq. 2).



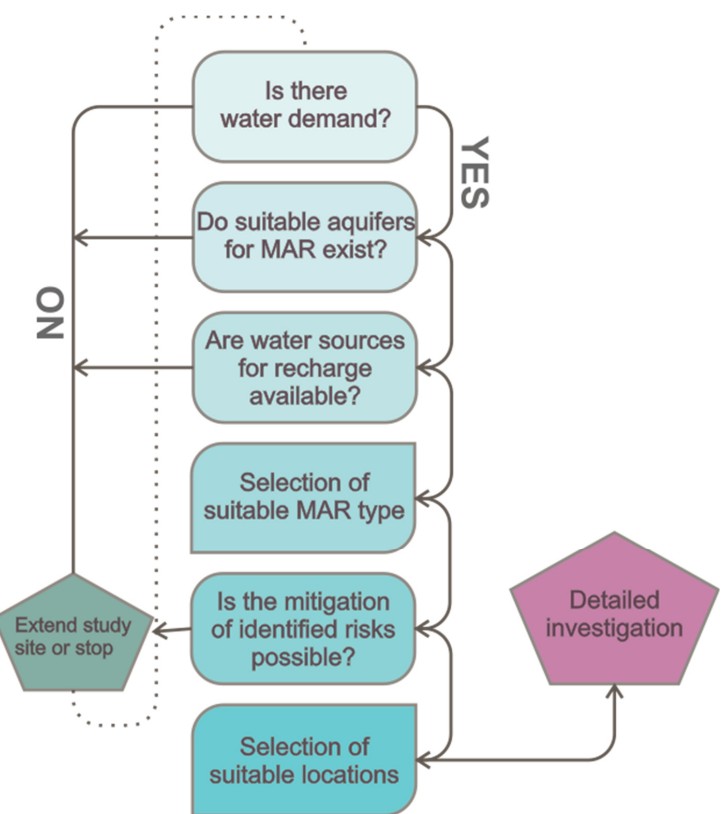

**Figure 3. Six-step decision tree for assessing the technical feasibility of managed aquifer recharge on the island of Grand Bahama.**

**3. Results and discussion**

**3.1 Part 1: Technical feasibility of managed aquifer recharge**

Following the six identified steps for the technical feasibility assessment (Figure 2), water demand was calculated to be 11,356 m³/d (3 mgd), corresponding to 30% of the currently brackish water supplied on the island of Grand Bahama. In a preliminary assessment, the recovery time of the aquifers by rainfall recharge was predicted to be 20 years (Al Baghdadi, 2021). A detailed

groundwater model to predict the recharge and groundwater flow needed to mitigate the saltwater intrusion of the brackish aquifer (by dilution) could not be prepared because of limited data for the study site. A requirement is also to identify aquifers


with adequate hydrogeological properties for storing and transmitting sufficient volumes of water. The entire island consists of karstified carbonates, and the latest available measurements document a porosity of 15-25% and hydraulic conductivities up to 2100 m/d, with strong variations due to the heterogeneity of the aquifers (Whitaker and Smart, 1997; Whitaker and

Smart, 2000). Due to the lack of detailed investigations of the karst system (e.g., caves or conduits, porous rock facies) on the island we assumed that generally the aquifers of the island could be suitable for MAR.

Rainwater was evaluated to be the most likely water source for a MAR scheme. Since surface water is not available on the island. Additionally, a major part of wastewater is treated locally in pit latrines and already recharge the aquifer. Analysis of rainfall data available from 2012 to 2022 revealed substantial precipitation amounts of 1594 mm/yr in average. Based on the

limited water source and the aquifers available on the island, rainwater harvesting was identified as the most suitable MAR type. The harvesting of rainwater in the Wellfields 1, 3 and 4 could be performed with rooftop rainwater harvesting and infiltration onsite into the aquifer via drain tranches installed locally on the properties. An evaluation of the proposed rainwater harvesting scheme with a drain trench was conducted in a risk assessment. Hazards were identified and a qualitative risk analysis and evaluation were conducted. The major human health risks identified were infiltration of saltwater or water with

high pollutant load during storm events into the drain trenches. Further, bird feces from rooftops can infiltrate, causing a microbiological contamination of the water. From a technical perspective, groundwater flooding due to an elevated groundwater table was identified as the major risk (detailed results of the risk assessment can be found in Section S3).

Based on the prior results, the following criteria for the selection of the most suitable MAR location were defined that also allow risk mitigation: (i) a minimum distance of the drain trench to the groundwater table to ensure sufficient natural treatment

of infiltrating water (purification within the unsaturated zone) and avoid groundwater flooding, (ii) a sufficiently high elevation against high storm surges, (iii) the use of rooftops for rainwater harvesting, where the location of MAR should be within a populated area. Furthermore (iv), the rainwater harvesting schemes should be located at suited areas that allow effective groundwater recharge. Suited areas with respect to the groundwater level (depth to the groundwater table) were mapped with an MCDA-GIS approach for Wellfields 1, 3 and 4 (Figure 4).



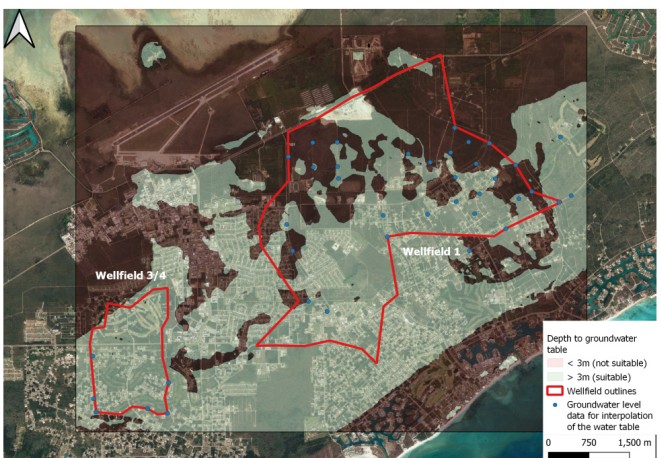


**Figure 4. MAR suitability map with respect to the depth to the groundwater table, for the populated areas of Wellfields 1, 3 and 4 (source of base map: © Google Maps)**

Available recharge volume from rooftop rainwater harvesting (RRWH) was estimated according to DIN (2002) based on the

roof area of the 2,456 buildings and average rainfall volumes for potential infiltration in Wellfield 1, Wellfield 3 and 4, and all

three wellfields together (Table 1). To obtain the surplus of recharge from the MAR rainwater harvesting scheme with a drain

trench, the current natural estimated recharge of 25 % (Little et al., 1977; Whitaker and Smart, 1997) was deducted from the

estimated recharge with a MAR scheme.

**Table 1. Roof area, recharge volume from rainwater harvesting, and resulting surplus recharge for the study areas**

| Study area | Roof area [m²] | Recharge volume from rainwater harvesting [m³] | Resulting surplus recharge [m³] |
|---|---|---|---|
| Wellfield 1 | 489,808 | 562,143 | 366,955 |
| Wellfields 3, 4 | 83,725 | 96,090 | 62,726 |
| Wellfields 1, 3, 4 | 573,533 | 658,503 | 429,681 |

A total of 429,681 m³/yr of additional recharge could be achieved with RRWH in Wellfields 1, 3 and 4, corresponding to

10.4% of the water demand for replacing the supply by brackish water. Therefore, the MAR scheme is not able to fully supply

the water demand on Grand Bahama, but rather contribute to a sustainable groundwater management practice. Unless an

investigation is conducted to identify groundwater flow paths, a reliable prediction of enhanced groundwater recharge originating from MAR rooftop rainwater harvesting schemes with drain trenches outside of the Wellfield 1, 3 and 4 is not reliable. From a technical perspective, the implementation of the RRWH in the 2,456 buildings in the wellfields would be possible. However, the construction of 2,456 RRWH schemes would be a time-consuming task, public acceptance would be a prerequisite to install these schemes on private terrain and the question who would take over the costs for the RRWH schemes

would need to be discussed.

### 3.2 Part 2: Financial CBA

### 3.2.1 Identification of the reforestation scenario

Results of the financial CBA and the extended CBA are presented in Table A1-A3 (Appendix A). Based on experts' opinion from Bahamas Forestry Unit (I. Miller, personal communication, February 23, 2022), the reforestation scenario comprehends

three areas (Figure 5): the first area (56.04 km$^2$) is located in Wellfield 6, where all mature pine trees were destroyed during Hurricane Dorian (Welsh et al., 2022); the second (70.30 km$^2$) and third areas (53.63 km$^2$) occupy public land in the East GB Forest Reserve, where Hurricane Dorian also affected the pine trees.

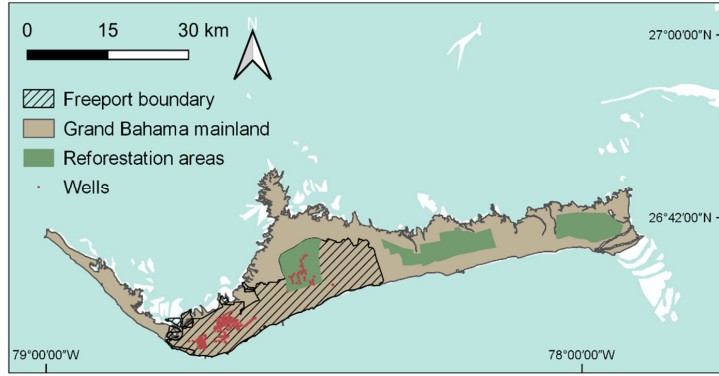

**Figure 5. Reforestation measure tested in Grand Bahama. EPSG: 4326.**

Based on the amount of fallen timber, one option for recovering costs could be for the land managers to obtain a permit to harvest. In addition, it was assumed that reforestation considers sustainable harvesting that does not affect future timber yields.



### 3.2.2. Net Present Value according to the financial CBA

The financial CBA included the supply of drinking water as a monetary benefit. The results of the estimated investment costs,
operation costs, and revenues were discounted over the 30-year period from 2020 to 2050 (Table 2). The planned water capacity
of the reverse osmosis (RO) system is 11,356 $m^3$/d (GBUC, 2021) and that of the RRWH measure is 1,177 $m^3$/d (Table A1).
With regards to RO, the investment costs are provided as a lumped sum. The operation costs of RO include variable and fixed
operation and maintenance costs (O&M) and annual repair and replacement (R&R) fund (Section S4 of the SI for see a detailed
description of the cost's estimation). The RO measure results for the financial CBA in a positive NPV of 51,131,907 USD
over a 30-year period with a discount rate of 4%.

The total investment cost of RRWH systems is estimated at 19.58 million USD, based on detailed project management and
administration, preparation of project, implementation of works and equipping, and promotion and viability costs (a detailed
description of the cost's estimation is provided in Section S5 of the SI). The implementation of the system in all 2,456 buildings
of Wellfields 1, 3, and 4 would require at least two years, leading to a longer time for investment costs. As a result of the
financial CBA for the RRWH measure, the NPV for a 30-year period at a discount rate of 4% is negative and equal to -
15,638,010 USD.

The total investment costs of the reforestation scenario are estimated at 103.89 million USD, based on detailed project
management and administration, preparation of project, implementation of works and equipping, and promotion and viability
costs (detailed description of the cost's estimation is provided in the Supplementary Material Section S6 of the SI). As
mentioned in the methodology, the reforestation measure in the financial analysis sees no estimations in terms of water supply,
leading to zero revenues. As a result of the financial CBA for the reforestation measure, the NPV for a 30-year period at a
discount rate of 4% is negative and equal to -135,690,081 USD.

Table 2 reports the results of the financial CBA in terms of NPV for the comparison of RO, MAR (RRWH), and reforestation
measures for multiple discount rates. When only drinking water supply is considered, RO is the best performing measure, with
positive NPV, increasing as the discount rate values get lower. The second-best performing measure according to the financial
CBA is the RRWH system, with negative NPV values, which increase as the discount rate value decreases. The worst-
performing measure in terms of water provisioning is reforestation, with negative NPV values, which increase proportionally
to the discount rate.

**3.3 Part 3: Extended CBA**

The extended CBA took into account as benefits not only the supply of drinking water, but also other ES (Table A1-A3). The
extended CBA for RO considered as benefits the drinking water supply and tourism, because these ES are based on water
capacity improvement. The RO measure results for the extended CBA in a positive NPV of 67,748,586 USD over a 30-year
period with a discount rate of 4%.

Similarly, to RO, the drinking water supply and tourism benefits were included as revenues for the potential MAR project, leading however to a negative NPV of -13,194,905 USD over a 30-year period with a discount rate of 4%. Instead, for reforestation, the carbon sequestration, habitat quality, and timber production benefits were included as revenues for the project, leading to a positive NPV of 71,879,831 USD for the 30-year period and discount rate of 4%.

Table 3 represents the results of the extended CBA for the tested measures for a set of ten discount rates from 1% to 10%.

When additional ES are considered, RO shows always positive NPV values, RRWH shows still negative NPV values, and the reforestation measure leads to mixed results in terms of profitability. Moreover, in comparison to a financial CBA, RO is not the best performing measure for all discount rate values anymore: for discount rate $r < 4\%$, reforestation shows higher NPV than RO; for $4\% \leq r \leq 7\%$, reforestation shows lower NPV than RO but still positive; for $r > 7\%$, the reforestation measure is not profitable.

It can be observed that, in all analyses, the discount rate has a big impact on the results. For five out of six measures, the lower the discount rate, the higher the NPV. This is explained by the fact that lower discount rates renumerate future benefits more than high discount rates (Martínez-Paz et al., 2014). Consequently, with low discount rates we see that environmental measures are more profitable because these see long term benefits. For the same principle, with high discount rates, the later the costs of a measure take place, the more profitable that measure will be. However, for the case of the reforestation measure in the

financial CBA (Table 2), the NPV is declining with the discount rate since only costs are considered. Researchers suggested different ways to deal with the uncertainty related to the discount rate: from using low discount rates for environmental projects (Costanza et al., 2017) to using multiple values according to time or service (Hanley, 2013; Martínez-Paz et al., 2014).

**Table 2. Net present value (NPV) of the financial CBA for the sustainablility measures reverse osmosis (RO), rooftop rainwater harvesting (RRWH), and reforestation. Project lifetime: 30 years. Yellow colors indicate low NPV, blue colors indicate high NPV.**

| Discount rate | NPV - RO [USD] | NPV - RRWH [USD] | NPV -Reforestation [USD] |
|---|---|---|---|
| 1% | 81,770,741 | -10,558,929 | -141,152,662 |
| 2% | 69,469,392 | -12,791,232 | -139,286,361 |
| 3% | 59,411,829 | -14,433,451 | -137,466,041 |
| 4% | 51,131,907 | -15,638,010 | -135,690,081 |
| 5% | 44,268,877 | -16,516,102 | -133,956,935 |
| 6% | 38,542,056 | -17,149,245 | -132,265,124 |
| 7% | 33,731,878 | -17,597,511 | -130,613,238 |
| 8% | 29,665,662 | -17,905,408 | -128,999,925 |
| 9% | 26,206,867 | -18,106,100 | -127,423,896 |
| 10% | 23,246,950 | -18,224,443 | -125,883,914 |





**Table 3. Net present value (NPV) of the extended CBA for the sustainability measures reverse osmosis (RO), rooftop rainwater**
**harvesting (RRWH), and reforestation. Project lifetime: 30 years. Yellow colors indicate low NPV, blue colors indicate high NPV.**

| Discount Rate | NPV - RO [USD] | NPV - RRWH [USD] | NPV -Reforestation [USD] |
|---|---|---|---|
| 1% | 107,481,410 | -6,178,826 | 203,119,863 |
| 2% | 91,531,821 | -9,224,216 | 148,247,921 |
| 3% | 78,488,806 | -11,497,203 | 105,463,219 |
| 4% | 67,748,586 | -13,194,905 | 71,879,831 |
| 5% | 58,843,897 | -14,461,647 | 45,343,367 |
| 6% | 51,411,177 | -15,403,621 | 24,237,241 |
| 7% | 45,166,027 | -16,099,374 | 7,342,061 |
| 8% | 39,884,776 | -16,607,308 | -6,266,867 |
| 9% | 35,390,561 | -16,971,094 | -17,294,565 |
| 10% | 31,542,787 | -17,223,574 | -26,281,533 |

### 3.4 Evaluation of the methodological aspects

#### 3.4.1 Technical feasibility of MAR

Technical feasibility studies for MAR measures are numerous and often apply common selection criteria or workflows (Sallwey et al., 2019). However, selection criteria must be adjusted based on regional or local (hydro-)geology. Hejazian et al. (2017b) investigated MAR implementation on an atoll in Marshall Islands, but did not include selection criteria used for their evaluation. Apart from the Marshall Islands study, no methodologies were available for MAR feasibility assessments on islands with freshwater lenses (FWLs). Hence, we needed to develop a new methodology including selection criteria. As a result, the
methodology was only applied for our study case and has not been successfully applied to the small island setting with freshwater lenses.

Moreover, the methodology applied on Grand Bahama had to be tailored to an investigation with scarce data availability. Similarly, Dobhal et al. (2019) suggested a methodology for river bank filtration with lower data availability in India. In their publication, selection criteria were not manifested with quantitative measures but rather with qualitative definitions. Further
research could improve availability of hydrogeological data, and the MAR potential could be explored with a methodology following a quantitative approach.

#### 3.4.2 Financial and extended CBA

The presented results with regards to the profitability of the measures depend on the methodological approach used for their estimation, as it still not agreed upon the most appropriate methodologies to assess monetary values of ecosystem services and
to include these estimates in a CBA. The CBA method has been widely applied in ecological restoration (Cerulus, 2014; Feuillette et al., 2016; Grossmann, 2012; Logar et al., 2019) and in water system assessments (Acuña et al., 2013; Ghafourian





et al., 2021; Ruangpan et al., 2020). However, this approach shows some limitations related to the lack of a method to estimate some benefits (Ruangpan et al., 2020), the estimation of the costs, the overuse of qualitative data, and the lack of validation of the results (Network Nature, 2022; Sudmeier-Rieux et al., 2021; Wegner and Pascual, 2011).

As for the reproduction of natural phenomena through modeling, the methods applied for the ecosystem services estimations are affected by uncertainties, e.g., with regards to the input data used (usually not location-specific) or concerning the parameterization of the models. A recommendation would be to consider at least two benefits per ecosystem function (Boithias et al., 2016), which is a biophysical relationship that exists regardless of whether or not humans benefit from an ecosystem (Costanza et al., 2017). For example, the ecosystem function of water storage and retention can contribute to the service

(benefit) of drinking water supply. Another recommendation after Boithias et al. (2016) would be to use multiple valuation metrics to value each benefit, e.g., applying willingness to pay and carbon prices to evaluate the benefit of carbon sequestration. Further research could consider alternative economic assessment models. For example, Wegner and Pascual (2011) suggest a pluralist framework of CBA composed of a heterogeneous set of value-articulating instruments, appropriate to the context within a specific decision. Saarikoski et al. (2016) state that multi-criteria decision analysis (MCDA) performs better than a

CBA because it allows including non-monetary ecosystem services. However, Perosa et al. (2022) showed the still existing limitations for the actual application of MCDA for river basin management. A combination of CBA and MCDA methods could be a potential solution (Saarikoski et al., 2016), although it implies higher efforts.

Besides methodological aspects related to the CBA itself, the absence of some ecosystem services in the extended CBA also represents a limitation, as it implies that the value of these ecosystem services has been set to zero. A first example of omitted

benefits are cultural services. InVEST provides a model ("Visitation: Recreation and Tourism") to estimate the effects of land use changes to nature-based recreation and tourism. The model uses the quantity and location of photos uploaded on Flickr to understand how landscape characteristics correlate to recreation and tourism. Within the framework of this publication, this InVEST model was tested but could not provide statistically significant results. A potential alternative to estimate nature-based recreation benefits could be to use the Travel Cost Method, for which a valid approach is suggested under the TESSA toolkit

(Peh et al., 2017). The method involves collecting data through interviews, which can be conducted in person or online, as done by Perosa et al. (2021) through social media.

Another example of ecosystem services (ES) not included in our study is crop pollination. InVEST provides a model for this ES as well, designed to model characteristics of nesting and foraging habitats of wild bees. As other factors influence pollination on Grand Bahama, the model was not applicable for this study area. Further research may use other models that

include different animal pollinators, or other weather drivers (e.g., wind) as another pollination factor.

Besides missing ecosystem services, other aspects are still missing from the CBAs of the analyzed measures. First, a potential additional benefit of MAR is that its implementation would decrease the water volume filtered by the RO system, with consequent energy savings. Second, a potential negative effect of the reforestation measures is that, most likely, these measures could decrease water recharge at the local level, which could affect groundwater lenses.





### 3.5 Services and costs generated by MAR compared to RO

For the investigated MAR schemes, only about 10.4% of the water demand could be supplied, whereas RO could supply 100% of the water demand. The financial and ES assessments showed that the MAR scheme would also be less profitable from a financial and an ES-based extended CBA.

A major difference between MAR and RO lies in the investment costs (5 million USD for RO compared to 22 million USD for MAR). The costs of the RO, which desalinates the groundwater but does not restore the aquifer, are lower than the costs of the MAR measure, which acts as an ecosystem restoration. However, this difference is not represented by the ecosystem service of water supply, which does not distinguish between water supplied from the RO plant or from the aquifer. An important improvement of this analysis would require finding a way to estimate the ecosystem service of aquifer recharge in addition to the ecosystem service of water supply.

### 3.6 Reforestation

The results of the financial and extended CBA for the reforestation measure indicated profitable results. The reforestation of pine forest would increase 10% of local stored carbon compared to current land use and carbon sequestration would generate 271 million USD along the analysis period of 30 years. Still, as discussed in Section 3.4, the results are subject to limitations related to data shortage.

Nevertheless, additional benefits could be generated by the reforestation planned in Wellfield 6 (Figure 4). Positive impacts on groundwater quantity and quality by forests were identified in that area, hence the reforestation measure could be implemented as a groundwater management strategy (Ellison, 2018). A known positive effect of the pine forest is the potential of phytoremediation, where salt is taken up by the plant and removed from the groundwater. However, vegetation might also cause the decrease of freshwater lenses (FWLs). Hejazian et al. (2017) studied an atoll in the Marshall Islands that consists of two lobes of land underlain by FWLs. One lobe was cleared from tropical forest due to military use and consequently, the FWL grew significantly in thickness due to a reduction of evapotranspiration. We recommend studying the effect of the forest on FWLs also on Grand Bahama. Furthermore, a potential benefit of reforestation is the increase of nature-based recreation caused by increased biodiversity, among others through birdwatching, one of the most popular tourist attractions on Grand Bahama.

### 3.7 Suggestions for sustainable groundwater management on Grand Bahama

Even after the implementation of the RO scheme, the population indicates insufficient desalinated water from Wellfield 6 to their households or even water outages (personal communication with population). The Grand Bahama Utility Company explains these shortcomings with problems pumping water from Wellfield 6 in sufficient quantity, likely because of lacks in the supply system. In comparison to the RO system, utilizing the MAR scheme of RRWH in Wellfields 1, 3, and 4 would





likely not receive an additional water load and would not strain the water supply system. The existing water supply infrastructure would be able to convey 10.4 % more water to the households. Therefore, the implementation of RRWH schemes should be considered as an additional option to provide a reliable water supply on the island, potentially also combining it with RO. Investigations of MAR feasibility should be reassessed after collection of (hydro-)geological information outside the wellfields. Adverse effects of the RO scheme such as high energy consumption or brine waste are further negative points of

its application. Nevertheless, the implementation of the RRWH scheme relates to long construction time and would require public acceptance for building such schemes on private premises.

Further measures such as the reduction of water use or the reduction of leakage losses, which currently account to 30-40% of the water demand, could be inspected (CDM, 2011). It is crucial that the public is involved in the decision making for groundwater and forest ecosystems restoration measures to gain acceptance for their implementation (UNEP, 2021). Relying

on the RO scheme as the only contingency plan for safe water supply on GB may be shortsighted.

## 4. Summary and conclusions

The Bahamas suffers from the consequences of recurring hurricanes. To mitigate these effects and restore the natural ecosystems, multiple measures have been discussed among stakeholders on Grand Bahama. Two planned sustainability measures, MAR and reforestation, were investigated for the mitigation of impacts of Hurricane Dorian on the island of Grand

Bahama. A holistic analysis of the two measures was conducted: an economic assessment was performed with a financial CBA, and the ecosystem services of the measures were investigated with an extended CBA. The existing RO scheme on the island was also assessed with the financial and extended CBA, and results were compared to the planned MAR measure for drinking water supply.

The proposed MAR scheme of rooftop rainwater harvesting with a drain trench from buildings in Wellfield 1, 3, and 4 on

Grand Bahama was technically evaluated and judged feasible. Nevertheless, the financial CBA evaluated the MAR scheme less profitable compared to the RO measure, which is explained by the difference in investment costs (22 million USD compared to 5 million USD for RO). Both the financial and the extended cost-benefit analysis methods do not distinguish between the two different ways in which RO and MAR supply freshwater, but accounts for a comparable ecosystem service: the freshwater supply. This leads to disregarding the additional value of the MAR scheme of regenerating the groundwater

ecosystem in comparison to a mere water supply provided by the RO system. We suggest that researchers investigate this aspect of MAR's benefits in the future. Areas for reforestation efforts were identified. The reforestation measure was assessed to be financially profitable and showed extensive potential to sustain the forest ecosystem services on the island.

The main limitation for the technical feasibility assessment of MAR on the island was a lack of hydrogeological data. We suggest further (hydro-)geological data collection outside of the wellfields and to reevaluate the MAR potential based on such

newly collected information. The financial CBA and extended CBA been criticized in the past with regards to how costs are





estimated, how benefits are modeled and monetized, and on the way how results are validated. Finally, obtained results within this study are subject to uncertainty due to the lack of detailed input data for the models and the assessments. Implementation of the sustainability measures on Grand Bahama is judged likely for the reforestation schemes.

The results of this work show that substantial financial and labor efforts are necessary to restore the forest and groundwater ecosystem on the island. Furthermore, this study supports that only a technical, economic, or ecological assessment of a planned human intervention in an environmental system falls short of accurately estimating its feasibility and benefit for the study area and its population. Therefore, a holistic approach considering different aspects should be pursued. The lack of data for MAR feasibility evaluation and extended CBA (financial assessment of nature-based solutions) effects obtained results and related uncertainty. Methods for technical, economic and ecosystem service assessments should be developed further in

the future to help decision makers in reaching the Sustainable Development Goals set by their governments.





**Appendix A**

**Table A1. Costs and revenues for the financial and extended cost-benefit analysis (CBA) of the RO measure along the 30-year analysis period, where the extended CBA is represented in the "Extended revenues" section. O&M: operation and maintenance. R&R: repair and replacement. Benefits related to the extended cost-benefit analysis are shown in the lower part of the table (red).**

| Description | Total Years 1 to 30 [USD] | Year 1 [USD] | Year 2 [USD] | Years 3-30 [USD·yr⁻¹] |
|---|---|---|---|---|
| **INVESTMENT COSTS** | | | | |
| Installation costs (surveys, studies, design, engineering) | 5,000,000 | 5,000,000 | | |
| Replacement cost | | | | |
| Residual value | | | | |
| **Total investment costs** | **5,000,000** | **5,000,000** | | |
| **OPERATION COSTS** | | | | |
| Fixed O&M | 1,401,667 | | 48,333 | 48,333 |
| Variable O&M | 2,682,500 | | 92,500 | 92,500 |
| Annual R&R | 6,670,000 | | 230,000 | 230,000 |
| Total operating costs | **10,754,167** | | 370,833 | 370,833 |
| **REVENUES** | | | | |
| Drinking water supply | 119,626,717 | | | 4,024,129 |
| **EXTENDED REVENUES (Extended CBA)** | | | | |
| Tourism | 35,794,517 | | | 1,078,589 |
| Total revenues | **155,421,234** | | | 5,102,719 |





**Table A2. Costs and revenues for the financial and extended cost-benefit analysis (CBA) of the MAR measure along the 30-year analysis period, where the extended CBA is represented in the "Extended revenues" section. Benefits related to the extended cost-benefit analysis are shown in the lower part of the table (red).**

| Description | Total Years 1 to 30 [USD] | Year 1 [USD] | Year 2 [USD] | Years 3-23 [USD·yr⁻¹] | Year 24 [USD] | Year 25 [USD] | Years 26-29 [USD·yr⁻¹] | Year 30 [USD] |
|---|---|---|---|---|---|---|---|---|
| **INVESTMENT COSTS (financial CBA)** | | | | | | | | |
| | | | | | | | | |
| **1. Project management and administration** | **7,638,730** | | | | | | | |
| Project manager | 77,400 | 23,220 | 54,180 | | | | | |
| Project administrator | 32,250 | 9,675 | 22,575 | | | | | |
| Experts in the installation of the system - Wellfield 1 | 6,733,600 | 2,020,080 | 4,713,520 | | | | | |
| Experts in the installation of the system - Wellfields 3/4 | 694,080 | | 694,080 | | | | | |
| Coordinator of works | 21,500 | 6,450 | 15,050 | | | | | |
| Financial manager | 41,200 | 12,360 | 28,840 | | | | | |
| Certificated expert for public procurement | 38,700 | 11,610 | 27,090 | | | | | |
| **2. Preparation of project** | **259,200** | | | | | | | |
| Water quality analysis | 19,200 | 19,200 | | | | | | |
| Study documentation | 144,000 | 144,000 | | | | | | |
| Project documentation | 64,000 | 64,000 | | | | | | |
| Permits obtaining | 32,000 | 32,000 | | | | | | |
| **3. Implementation of works and equipping** | **18,471,926** | | | | | | | |
| Self-cleaning filter - Wellfield 1 | 357,258 | 107,177 | 250,081 | | | | | |
| Self-cleaning filter - Wellfields 3/4 | 50,453 | | 50,453 | | | | | |
| Gutter system - Wellfield 1 | 6,481,957 | 1,944,587 | 4,537,370 | | | | | |
| Gutter system - Wellfields 3/4 | 269,360 | | 269,360 | | | | | |
| Distribution piping - Wellfield 1 | 232,575 | 69,773 | 162,803 | | | | | |
| Distribution piping - Wellfields 3/4 | 25,305 | | 25,305 | | | | | |
| Excavation of soakaway - Wellfield 1 | 9,967,500 | 2,990,250 | 6,977,250 | | | | | |
| Excavation of soakaway - Wellfields 3/4 | 1,084,500 | | 1,084,500 | | | | | |
| Gravel for soakaway - Wellfield 1 | 2,577 | 773 | 1,804 | | | | | |
| Gravel for soakaway - Wellfields 3/4 | 441 | | 441 | | | | | |
| **4. Promotion and visibility** | **19,394** | | | | | | | |
| Ad campaign | 19,394 | 9,697 | 9,697 | | | | | |
| Initial investment (1+2+3+4) | 26,389,251 | 7,464,852 | 18,924,398 | | | | | |
| **5. Replacement cost** | **6,751,317** | | | | | | | |
| Gutter replacement - Wellfield 1 | 6,481,957 | | | | 3,240,979 | 3,240,979 | | |



| Description | Total Years 1 to 30 [USD] | Year 1 [USD] | Year 2 [USD] | Years 3-23 [USD·yr⁻¹] | Year 24 [USD] | Year 25 [USD] | Years 26-29 [USD·yr⁻¹] | Year 30 [USD] |
|---|---|---|---|---|---|---|---|---|
| Gutter replacement - Wellfields 3/4 | 269,360 | | | | | 269,360 | | |
| **6. Residual value** | **-13,603,026** | | | | | | | -11,337,988 |
| **Total investment costs** | **21,802,580** | **7,464,852** | **18,924,398** | | **3,240,979** | **3,510,339** | | **-11,337,988** |
| **OPERATION COSTS (financial CBA)** | | | | | | | | |
| System maintenance (monthly fee) | 168,000 | | | 6,000 | 6,000 | 6,000 | 6,000 | 6,000 |
| Experts in replacement of gutters - Wellfield 1 | 1,683,400 | | | | 841,700 | 841,700 | | |
| Experts in replacement of gutters - Wellfields 3/4 | 183,160 | | | | | 183,160 | | |
| Regular water quality analysis | 860,160 | | | 30,720 | 30,720 | 30,720 | 30,720 | 30,720 |
| Total operating costs | **2,894,720** | | | 36,720 | 878,420 | 1,061,580 | 36,720 | 36,720 |
| **REVENUES (financial CBA)** | | | | | | | | |
| Drinking water supply | 16,901,344 | | | 603,619 | 603,619 | 603,619 | 603,619 | 603,619 |
| **EXTENDED REVENUES (extended CBA)** | | | | | | | | |
| Increase in tourism | 3,171,052 | | | 113,252 | 113,252 | 113,252 | 113,252 | 113,252 |
| Total revenues | **20,072,396** | | | 716,871 | 716,871 | 716,871 | 716,871 | 716,871 |





**Table A3. Costs and revenues for the financial and extended cost-benefit analysis (CBA) of the reforestation measure along the 30-year analysis period, where the extended CBA is represented in the "Extended revenues" section. Benefits related to the extended cost-benefit analysis are shown in the lower part of the table (red).**

| Description | Total Years 1 to 30 [USD] | Year 1 [USD] | Year 2 [USD] | Year 3 [USD] | Year 4 [USD] | Years 5-29 [USD·yr⁻¹] | Year 30 [USD] |
|---|---|---|---|---|---|---|---|
| **INVESTMENT COSTS (financial CBA)** | | | | | | | |
| **1. Project management and administration** | **211,050** | | | | | | |
| Project manager | 77,400 | 23,220 | 54,180 | | | | |
| Project administrator | 32,250 | 9,675 | 22,575 | | | | |
| Coordinator of works | 21,500 | 6,450 | 15,050 | | | | |
| Financial manager | 41,200 | 12,360 | 28,840 | | | | |
| Certificated expert for public procurement | 38,700 | 11,610 | 27,090 | | | | |
| 2. Preparation of project | 240,000 | | | | | | |
| Study documentation | 144,000 | 144,000 | | | | | |
| **Project documentation** | **64,000** | 64,000 | | | | | |
| Permits obtaining | 32,000 | 32,000 | | | | | |
| 3. Implementation of works and equipping | 103,421,822 | | | | | | |
| Site preparation | 4,391,367 | 4,391,367 | | | | | |
| Pre-planting site survey | 235,130 | 235,130 | | | | | |
| **Tree planting** | **64,130,690** | 42,208,216 | 21,922,474.44 | | | | |
| Materials and equipment | 6,887,124 | 4,565,870 | 2,321,254 | | | | |
| Labor | 25,555,921 | 15723288.93 | 9,832,632 | | | | |
| Transportation | 2,221,589 | 1,790,908 | 430,681 | | | | |
| 4. Promotion and visibility | 19,394 | | | | | | |
| Ad campaign (newspaper, television and radio) | 19,394 | 9,697 | 9,697 | | | | |
| Initial investment | 103,892,266 | 69,227,792 | 34,664,474 | | | | |
| **5. Residual value** | | - | - | | | | |
| **Total investment costs** | **103,892,266** | **69,227,792** | **34,664,474** | | | | |
| **OPERATION COSTS (financial CBA)** | | | | | | | |
| Maintenance | 37,641,086 | 25,156,543 | 12,484,543 | | | | |
| Monitoring | 1,533,291 | | | 974,880 | 558,411 | | |
| Total operating costs | **39,174,376** | 25,156,543 | 12,484,543 | 974,880 | 558,411 | 0 | 0 |
| **REVENUES (financial CBA)** | | | | | | | |
| Revenues | - | - | - | - | - | - | - |
| **EXTENDED REVENUES (extended CBA)** | | | | | | | |
| Carbon sequestration | 270,853,348 | 9,028,445 | 9,028,445 | 9,028,445 | 9,028,445 | 9,028,445 | 9,028,445 |
| Habitat quality | 23,800,080 | 793,336 | 793,336 | 793,336 | 793,336 | 793,336 | 793,336 |
| Timber production | 122,377,765 | | | | | | 122,377,765 |
| Total revenues | **417,031,193** | **9,821,781** | **9,821,781** | **9,821,781** | **9,821,781** | **9,821,781** | **132,199,546** |



## Author contribution

Conceptualization, Imig, A., Perosa, F., Rein, A.; methodology, all authors; software, Perosa, F., Iwane Hotta, C., Klausner, S.; validation, all authors.; formal analysis, all authors; investigation, all authors; data curation, Iwane Hotta, C., Klausner, S.; writing—original draft preparation, Imig, A. and Perosa, F.; writing—review and editing, all authors.; visualization, Imig, A., Perosa, F., Klausner, S., Iwane Hotta, C.; supervision, Rein, A., Welsh, K., Perosa, F., Imig, A.; project administration, Rein, A., Welsh, K., Perosa, F., Imig, A; funding acquisition, Rein, A., Welsh, K., Perosa, F., Imig, A. All authors have read and agreed to the published version of the manuscript.

## Funding

This research was funded by Bahamas Protective Area Fund, grant number FU20210818 and, TUM Global Incentive Fund Call 11 and the BayIntAn agreement BAyIntAn_TUM_2022_23.

## Data Availability Statement

The result data set and the scripts used to create the data set of this study will be available at the mediaTum data repository (institutional repository of the Technical University of Munich) after acceptance (DOI: NN).

## Acknowledgments

We acknowledge the support from the Grand Bahama Utility Company for this research helping to understand the urban water supply system in the Bahamas. This work was partially funded through Bahamian protective area fund (BPAF) project SOFTGR, German academic exchange Service (DAAD) PROMOS fund, and the BayIntAn fund from the Bavarian Research Alliance.

## Competing interests

The funders had no role in the design of the study; in the collection, analyses, or interpretation of data; in the writing of the manuscript; or in the decision to publish the results. The authors declare that they have no conflict of interest.

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
