# Peer review of "Technical assessment combined with extended cost-benefit analysis for groundwater ecosystem services restoration - An application for Grand Bahama"

_Hydrology and Earth System Sciences, 2023_

## Referee Comment (RC1)

**Principle review criteria**

Scientific significance: 1 – Excellent

Scientific quality: 2 – Good

Presentation quality: 2 - Good

**General comment**

Dear authors,

The paper introduced a novel approach for combining a technical assessment and a cost benefit analysis for decisions on water supply solutions on an island. The cba is divided into a financial cba and an extended cba demonstrating the importance of accounting more than pure project internal financial consequences in decisions. The extended cba provides examples on how ecosystem services such as carbon sequestration can be monetized and thus included in a cba. The paper delivers an important and not widely covered topic of combining technical assessments into economic evaluation including monetizing effects on ecosystem services. The paper contributes to the state-of-the-art provided some amendments suggested below.

**Specific comments**

Row 83: The authors state, with references, that the CBA method falls short to adequately monetarize ecosystems services. I would prefer it if this was described with more details, thus, in what way does it falls short and why?

Row 91: Is it reasonable to say that the aim of the result is to show financial benefits of NBS? It sounds a bit biased. Maybe it would be better to say that the result from the CBA aims at providing a systematic review of different measure alternatives where NBS is one that is compared to more traditional ones? Thus, the result should speak for itself; the aim should not be to get a certain result.

Also, I think the aim could benefit from having a few objectives as well specifying more directly what has been carried out in the study…. E.g., 1) developing the technical assessment of MAR on tropical islands, 2) developing the methodology for extended cba with ES-analysis, 3) demonstrating the method on a case study etc…

Row 170 and forward: The methodology is not sufficient enough. The criteria is not defined or explained. One table naming all the parameters / criteria used with an explanation on what data, what tools and what criteria value that were used for the evaluation needs to be explained. The MCDA is not explained in sufficient details eighter. Please rewrite this part and provide sufficient information on the methodology so that the reader of the text has the possibility to judge the method and understand the procedure.

Row 195: I would like to have a comment of the chosen project time/ life time of the project. 30 years seems a bit short for a large project as a drinking water supply solution.

Section 2.4-2.5: In general, it is difficult to follow the procedures and what effects that are included in what CBA. as it is now, the reader must go back to the main text in the methodology in order to be able to interpret the result shown in the CBA-tables. This makes interpretating the result difficult and time consuming. I suggest a table that clarifies the differences between the three analyses in a structured way where a summary on what parameters are included in e.g., the financial cba compared to the extended cba for both the RO, RRWH, and reforestation.

Discussion/summary/conclusions: I think the discussion of the result could benefit greatly by including a thorough discussion about the uncertainties associated with the analysis. With uncertainties I mean:

- Are the parameter values, thus the numbers used in the CBA certain or could the costs and benefits differ?
- Are the models used to determine the values of the different cost and benefit items in the CBA certain, or could other models/other ways of valuating effect have an impact on the value of the cost-benefit item and thus the overall result?
- Are there effects that have not been included in the extended cba that could have had an impact on the overall result?

**Technical comments**

Row 113 and 115: You do not have to refer to figure 1 once again.

Row 152: figure 3?

Figure 3 could be moved closer to section 2.4 where it is referenced.

Result section: Wouldn't it be better to have the result tables located closer to the text describing the results?

---

## Referee Comment (RC2)

Principle review criteria

Scientific significance: 2 – Good

Scientific quality: 2 – Good

Presentation quality: 3 - Fair

**Review:**

The study provides a comprehensive analysis of the aftermath of Hurricane Dorian's impact on Grand Bahama Island, specifically addressing the extensive flooding and saltwater intrusion into aquifers, which significantly affected the island's water supply. Through an exploration of Managed Aquifer Recharge (MAR) and reforestation as potential nature-based solutions, the study conducts a thorough technical assessment of MAR, identifying plausible implementation sites. Additionally, it offers insightful financial and cost-benefit analyses, integrating ecosystem services, for both MAR and reforestation strategies.

The study's approach is noteworthy for its emphasis on holistic consideration and sustainability. While not exhaustive, it offers relevant implications for addressing urgent environmental challenges and enhancing the resilience of ecosystems and local communities in Grand Bahama.

**Comments**

- The abstract effectively outlines the problem, methodology, and results. However, to enhance clarity, it would be beneficial for the authors to explicitly state the purpose of the study and its specific objectives. Additionally, while the exploration of Managed Aquifer Recharge (MAR) is well-presented, the findings on reforestation could be given equal prominence to provide a comprehensive overview of the conservation practices studied.

- Line 51 contains an important fact that warrants citation.

- Line 58 highlights that the RO system fails to reach potable levels in certain households according to WHO standards. It would strengthen the discussion to specify the extent of this shortfall and explore potential reasons behind it, such as technological limitations or system deterioration. Providing a citation to the potability level guidelines would enable readers to verify this information.

- Line 60 mentions corroded pipes in Bahamian households, likely due to high water salinity. Is this linked to GB's RO system? Clarification is needed to understand its impact on system performance.

- In line 65, the authors could expand on whether RO systems have been previously utilized in the area and investigate evidence of hurricane damage leading to energy supply disruption in the study area or similar regions.

- Line 67 presents an important aspect regarding the benefits of Natural Based Solutions. It would be beneficial for the authors to support this assertion with recent references

- Lines 75 to 77 contain significant facts. It is recommended to accompany these facts with recent evidence and publications to strengthen the argument.

- In natural conditions, when rainfall occurs, the initial portion often infiltrates into the soil. When the soil becomes saturated, or if the rainfall intensity exceeds the infiltration capacity of the soil, excess water will indeed flow over the land surface as runoff, and occasionally causing flooding (Smith, R., & Goodrich, D, 2005. https://www.tucson.ars.ag.gov/unit/publications/PDFfiles/1696.pdf). In line 139, the paper suggests that excess rainfall can be infiltrated, contradicting or suggesting otherwise of what is stated above. Could you clarify this?

- The last paragraph of section 2.4 (Starting from line 132) is not clearly explained. Since the subsequent analysis relies on the identification of feasible MARs, this paragraph should be sufficiently elucidated. For instance, it is not clearly indicated how the water demand and suitable aquifers were determined. The article should be self-explanatory, and the supplementary material should be used as a complement. However, in this case, many fundamental criteria of the different steps of the methodology were placed in the supplementary material. It is recommended to rewrite this paragraph providing more detail of the methodology.

- The equation presented in Equation 2 exhibits a technical flaw: the absence of an opening bracket and the omission of summation limits. These oversights compromise the clarity and accuracy of mathematical expressions.

- Equation 3 introduces a subscript for the variable NPV, which is absent in Equation 2. It would be helpful to clarify the significance of these subscripts for NPV to ensure consistency and understanding throughout the equations.

- Line 197 mentions that the costs of RO were based on global reference costs from 2021, while operational costs were derived from literature predating 2018. To improve accuracy, consider incorporating more recent operational costs for a more precise assessment.

- The information provided in section 2.5 is limited and not well organized. For example, the five ecosystem services indicated are not addressed with the same proportion or depth, nor do they follow the indicated sequence. For instance, habitat provisioning (ecosystem service 4) is discussed after Timber provisioning (ecosystem service 5). Tourism (ecosystem service 2) is addressed at the end of the section.

- Navigating through the article proves challenging as it requires frequent backtracking to previous sections or referring to supplementary material to grasp the primary configurations that the article evaluates. Therefore, it is suggested to rewrite the text.

- On line 252, Figure 2 is cited, which supposedly involves six steps. However, it appears that this reference is incorrect, as the corresponding figure should be Figure 3.

- In line 252, the water demand is estimated to be 30% of the current supplied on the island. Shouldn't this demand be calculated based on the population instead?

  Furthermore, it would be valuable to discuss the correlation between the aquifer's recovery dynamics and the water demand necessary to fulfill a portion of the water supply. For instance, if it is anticipated that the aquifer will fully replenish in 20 years, it raises questions about the interplay between this recovery timeline and the water demand. If non-natural interventions are projected over a 30-year evaluation period but the aquifer recuperates within 20 years, there is a possibility that the populace might require less water treated through osmosis. Nevertheless, it's essential to consider that the population size might have increased by the 20-year mark as well.

- In line 263, it is indicated that a major part of wastewater is treated locally in pit latrines and already recharges the aquifer. Could the authors comment on whether this treatment is sufficient to avoid compromising the water quality of the aquifer? If these latrines are located near the proposed or existing extraction wells, could they compromise the water quality of the aquifer, thus requiring additional costs for treatment for drinking water purposes?

- Figure 3 could be improved for better information transfer. For example, the arrows do not clearly indicate the direction of the flow chart.

- In line 267, the article mentions "via drain tranches". It is important to clarify if this refers to drain trenches.

- Figure 4 is not self-explanatory. For instance, it is not possible to identify where this is located on the general map of Grand Bahama. What does the prominent rectangle signify? Is it the area of analysis? Where is the "<3m (no suitable)" area shown in the figure? The figure indicates that the blue dots represent groundwater level data. To what level does it refer? Distinguished values of the level are not apparent.

- In line 292, volumes of recharge are indicated in m3/yr, whereas in previous paragraphs, demand is indicated in m3/d. It is suggested to standardize the units for better communication of results to the readers.

- The final paragraph of section 3.1 appears to be somewhat confusing. Initially, it suggests that the implementation of RRWH would be technically feasible. However, it then mentions that the construction of schemes would be a time-consuming task, and that public acceptance would be a prerequisite. Could you please provide clarification on this matter?

- In line 336, the text highlights reforestation as the least effective measure for water supply. However, the correlation between reforestation and water supply is not explicitly addressed in the article. If this relationship is not defined, in this article reforestation serves no other purpose than ecosystem services. Therefore, its inclusion/comparison in CBA is questioned. Referring to what is mentioned in the article "The worst performing measure in terms of water provisioning is reforestation," could the author include the relationship between reforestation and water provisioning?

- The content of Section 3.4.2 provides valuable information on the criteria/methods used to estimate/assess the benefits of ecosystem services, as well as those services not addressed by the study, among others. Incorporating this information into the introduction could enhance the clarity of the article in understanding the general parameters of the study. It is suggested to include relevant details from this section in the introduction to illustrate to the reader why the study focuses on the applied methodology and the natural measures addressed.

- In line 440, the article references " Positive impacts on groundwater quantity and quality by forests were identified in that area…" However, it was not possible to verify/contrast the amounts of improvements in the quantity and quality of groundwater attributed to reforestation. It would be beneficial to present these results/findings in tables to provide a more detailed overview of the benefits of reforestation.

- The article considers various assumptions due to lack of data, such as using habitat provisioning data from Rio Grande Basin - Texas (Wang et al. (2021)), data based on communication with experts, and not having groundwater modeling, among others. The results and discussion should include an extended analysis of uncertainties associated with these sources.

- The article addresses an important topic of assessing Nature-based solutions to mitigate the impacts of Hurricane Dorian in Grand Bahama. However, it is suggested to reformulate the discussion and conclusions based on the comments and feedback provided.

---

## Author Comment (AC1)

**RC1: 'Comment on hess-2023-236', Anonymous Referee #1, 15 Feb 2024:**

**General comment**

Dear authors,

The paper introduced a novel approach for combining a technical assessment and a cost benefit analysis for decisions on water supply solutions on an island. The cba is divided into a financial cba and an extended cba demonstrating the importance of accounting more than pure project internal financial consequences in decisions. The extended cba provides examples on how ecosystem services such as carbon sequestration can be monetized and thus included in a cba. The paper delivers an important and not widely covered topic of combining technical assessments into economic evaluation including monetizing effects on ecosystem services. The paper contributes to the state-of-the-art provided some amendments suggested below.

REPLY: We would like to thank you for your time and work to review our manuscript. We are happy you found the work contributes to the state-of-the-art of combining technical assessments into economic evaluation including monetizing effects on ecosystem services. Please find below our answers to the specific comments.

**Specific comments**

Row 83: The authors state, with references, that the CBA method falls short to adequately monetarize ecosystems services. I would prefer it if this was described with more details, thus, in what way does it falls short and why?

REPLY:

We have extended the text in the manuscript as follows (changes/additions marked yellow):

"CBA analysis has been applied in existing literature to assess the economic feasibility of MAR projects (e.g., Halytsia et al., 2022; Rupérez-Moreno et al., 2017) but has not included ecosystem services (ES): one of the highlighted benefits of NBS. Furthermore, the CBA method falls short to adequately monetarize ecosystems services (e.g., Maliva, 2014; Ruangpan et al., 2020; Network Nature, 2022; Sudmeier-Rieux et al., 2021; Wegner and Pascual, 2011). In fact, by definition a CBA should be able to consider all benefits and costs of a measure by translating social, environmental and economic aspects into monetary values (Clinch, 2004; Hanley, 2013). Often, however, only partial benefits of a measure are included in a CBA, especially marketed values (Clinch, 2004), thereby neglecting ethical and cultural aspects (Vojinovic et al., 2017), and implicitly setting all neglected benefits to zero (Dominati et al., 2014). Therefore, we propose a methodology in this study that sets itself apart from already published research as it aims to combine a technical feasibility assessment and use the results to assess them in an extended cost-benefit analysis (CBA) with ecosystem services analysis. Ecosystem services are modelled with the InVEST software (Sharp et al., 2020)."

Additional references:

Clinch, J. (2004). Cost–benefit analysis applied to energy. In C. J. Cleveland (Ed.), Encyclopedia of energy (pp. 715–725). Elsevier Acad. Press. https://doi.org/10.1016/B0-12-176480-X/00237-0

Dominati, E. J., Robinson, D. A., Marchant, S. C., Bristow, K. L., & Mackay, A. D. (2014). Natural capital, ecological infrastructure, and ecosystem services in agroecosystems. In Encyclopedia of agriculture and food systems (pp. 245–264). Elsevier. https://doi.org/10.1016/B978-0-444-52512-3.00243-6

Hanley, N. (2013). Environmental cost–benefit analysis. In Encyclopedia of energy, natural resource, and environmental economics (pp. 17–24). Elsevier. https://doi.org/10.1016/B978-0-12-375067-9.00103-0

Vojinovic, Z., Keerakamolchai, W., Weesakul, S., Pudar, R., Medina, N., & Alves, A. (2017). Combining ecosystem services with cost-benefit analysis for selection of green and grey infrastructure for flood protection in a cultural setting. Environments, 4 (1), 3. https://doi.org/10.3390/environments4010003

Row 91: Is it reasonable to say that the aim of the result is to show financial benefits of NBS? It sounds a bit biased. Maybe it would be better to say that the result from the CBA aims at providing a systematic review of different measure alternatives where NBS is one that is compared to more traditional ones? Thus, the result should speak for itself; the aim should not be to get a certain result.

REPLY: We agree with this suggestion, we have reformulated the paragraph as follows: ”Results aim to allow a systematic comparison of NBC and RO costs and benefits to policy and decision makers and help justify their implementation.”

Also, I think the aim could benefit from having a few objectives as well specifying more directly what has been carried out in the study…. E.g., 1) developing the technical assessment of MAR on tropical islands, 2) developing the methodology for extended cba with ES-analysis, 3) demonstrating the method on a case study etc…

REPLY: We agree with the suggestion and have adapted the paragraph starting in row 88: ”In this work, (i) a technical assessment including risk assessment of MAR on small island nations is developed. Next, (ii), a methodology for an extended CBA with ES analysis is proposed. The methodology aims to explore the feasibility of NBS and RO from an economic and ecosystem services perspective. The two developed methodologies are then, (iii), applied to a study case on the island of Grand Bahama, The Bahamas. This study aims to show methods for investigating ecosystem services from an economic perspective. Results allow a systematic comparison of NBC and RO costs and benefits for, e.g., policy and decision makers and help justify their implementation.”

Row 170 and forward: The methodology is not sufficient enough. The criteria is not defined or explained. One table naming all the parameters / criteria used with an explanation on what data, what tools and what criteria value that were used for the evaluation needs to be explained. The MCDA is not explained in sufficient details eighter. Please rewrite this part

and provide sufficient information on the methodology so that the reader of the text has the possibility to judge the method and understand the procedure.

REPLY: We have extended this section, to explain needed data, the parameters/criteria, the tools applied and the criteria for the values better. Nevertheless, we think a display in a table is not suitable. As for the risk assessment, for the suitable aquifer and for the selection of the MAR scheme, the criteria vary based on site specific information collected and decisions taken in the steps before. Nevertheless, we have added a table with a general overview of the methodology (see below). Therefore, the applied tools and needed data base can only be narrowed down after following different selection steps (explained in the results section). For example, a review paper by Sallwey et al. (2019) has attempted to describe the summarized criteria used for the selection of suitable MAR locations. Another example is the review paper by Imig et al. (2022), where the authors summarize decision criteria and different methods used for risk assessment of MAR sites based on data availability. Therefore, we have restricted the methodology description to the site-specific information in Grand Bahama and added explanation about data, and decision criteria in the result section. We propose to extend section 2.4 as follows, after line 155:

"As the first step, the **(i) water demand** is defined, as without water demand a MAR scheme is not needed. The demand can either be defined based on technical guidelines from the country's legislation or based on the documented water use of the consumers. It is reasonable to predict a water demand for the design life of the MAR measure, e.g., commonly 30 years are set in water supply infrastructure. We collected data from the water authorities in Grand Bahama to calculate the water demand. For the **(ii) identification of suitable aquifers,** the hydrogeological properties of regional aquifers were collected. Hydrogeological properties should include the lithology and the location of the aquifer, storage capacity, and hydraulic conductivity (DEEPWATER-CE, 2020a; NRMMC, 2006). Based on the available data and the site-specific information, a suitable aquifer with sufficient storage capacity to supply the water demand shall be chosen. After defining the water demand and a suitable aquifer, the **(iii) water source(s) for groundwater recharge** should be identified, e.g., rainwater, surface water, or desalinated water. Based on the available water source, a **(iv) suitable MAR scheme** can be selected for the water demand and the available aquifers. This is necessary as, e.g., rainwater harvesting schemes have different requirements regarding groundwater levels compared to a riverbank filtration scheme (Sallwey et al., 2019). Specific criteria and data needed for their identification were determined in a literature review. They are not further summarized here but specified for the chosen MAR type in the results section. For step **(v),** we conducted a qualitative **risk assessment** with a risk score matrix after Swierc et al. (2005). Potential hazards for a MAR scheme in Grand Bahama for the risk assessment were chosen from a collection published in a review paper by Imig et al. (2022). For step **(vi) selection of suitable location,** we developed selection criteria based on information gained from the previous steps. Similar to step (iv), the selection criteria and the data we refrained from further specification in the methodology section for all possible MAR types due their quantity. The criteria for the chosen MAR scheme (in step (iv)) are summarized in the results section based on information from DEEPWATER-CE (2020a), CEHI et al. (2010), and NRMMC (2006). The criteria were assessed using the geographical information system QGIS

(2020) and were used in a multi-criteria decision analysis (MCDA) (Sallwey et al., 2019). The achievable recharge volume from the rainwater harvesting scheme was calculated based on recommendations by the German institute for norms (DIN, 2002), where details are given in Section S1 in the Supporting Information (SI). If the steps (i)-(iii) and (v) generate a negative evaluation, we suggest extending the study area or stopping the investigation. Otherwise, if all steps can be followed and result in a positive evaluation, MAR is considered to be feasible for the study site. Input data used to conduct the technical feasibility assessment (and the other parts of the holistic analysis) are described in Table S1 of the SI."

Publications mentioned in this reply:

Sallwey, J., Bonilla Valverde, J.P., Vásquez López, F., Junghanns, R., Stefan, C., 2019. Suitability maps for managed aquifer recharge: A review of multi-criteria decision analysis studies. Environ. Rev. 27, 138–150. https://doi.org/10.1139/er-2018-0069

Imig, A., Szabó, Z., Halytsia, O., Vrachioli, M., Kleinert, V., & Rein, A. (2022). A review on risk assessment in managed aquifer recharge. *Integrated Environmental Assessment and Management*, *18*(6), 1513–1529.

Row 195: I would like to have a comment of the chosen project time/ life time of the project. 30 years seems a bit short for a large project as a drinking water supply solution.

REPLY: We agree that the timeframe seems short, and maybe even not sustainable as drinking water will be needed also after that period. The timeframe of thirty years was chosen after communication with engineer offices such as Phoenix Engineer and personal experiences from prior work in the water supply infrastructure sector. Most water infrastructure projects are funded by governmental institutions and are influenced by current politics and public opinion. We think it lies in the nature of the funding sources that the projects are not planned for a longer period as the current government will not benefit from their effect if they have to calculate expenses extending over their legislative period.

Section 2.4-2.5: In general, it is difficult to follow the procedures and what effects that are included in what CBA. as it is now, the reader must go back to the main text in the methodology in order to be able to interpret the result shown in the CBA-tables. This makes interpretating the result difficult and time consuming. I suggest a table that clarifies the differences between the three analyses in a structured way where a summary on what parameters are included in e.g., the financial cba compared to the extended cba for both the RO, RRWH, and reforestation.

REPLY: We agree that the methodology could be complicated to understand at a first glance, and we thank you for the suggestion on adding a schematization of the methodology through a table. We have added the following table in the methodology, for better understanding, while reviewing the results.

| Factors | Technical feasibility | Financial CBA | Extended CBA | Analyzed measure |
|---|---|---|---|---|
| Water demand | ✓ | | | MAR types (incl. RRWH) |

| | | | | |
|---|---|---|---|---|
| Aquifer type | ✓ | | | MAR types (incl. RRWH) |
| Water source for MAR | ✓ | | | MAR types (incl. RRWH) |
| MAR technique | ✓ | | | MAR types (incl. RRWH) |
| Risk assessment | ✓ | | | MAR types (incl. RRWH) |
| Location | ✓ | | ✓ | MAR types (incl. RRWH) |
| Measure's costs (C) | | ✓ | ✓ | RRWH, RO, reforestation |
| Benefits of the drinking water supply (DWS) | | ✓ | ✓ | RRWH, RO, reforestation |
| ES of carbon sequestration (Carbon) | | | ✓ | RRWH, RO, reforestation |
| ES of timber provisioning (TP) | | | ✓ | RRWH, RO, reforestation |
| ES of habitat provisioning (HP) | | | ✓ | RRWH, RO, reforestation |
| ES of tourism (T) | | | ✓ | RRWH, RO, reforestation |

Discussion/summary/conclusions: I think the discussion of the result could benefit greatly by including a thorough discussion about the uncertainties associated with the analysis. With uncertainties I mean:

- Are the parameter values, thus the numbers used in the CBA certain or could the costs and benefits differ?
- Are the models used to determine the values of the different cost and benefit items in the CBA certain, or could other models/other ways of valuating effect have an impact on the value of the cost-benefit item and thus the overall result?
- Are there effects that have not been included in the extended cba that could have had an impact on the overall result?

REPLY: Thank you very much for outlining these topics. Please see our reply below, to a question on uncertainty raised by the second reviewer.

**Technical comments**

Row 113 and 115: You do not have to refer to figure 1 once again.

REPLY: We have removed the references to Figure 1.

Row 152: figure 3?

REPLY: We have moved the initial reference of Figure 3 from line 157 to line 152.

Figure 3 could be moved closer to section 2.4 where it is referenced.

REPLY: We have moved Figure 3 to the end of section 2.4.

Result section: Wouldn't it be better to have the result tables located closer to the text describing the results?

REPLY: We have added the extensive results tables to the Appendix to allow the reader to not be interrupted in the reading flow by the tables. We will contact the layouting team of HESS to discuss possibilities to display the tables in a published version and evaluate their location again.

**RC2: 'Comment on hess-2023-236', Anonymous Referee #2, 16 Apr 2024**

The study provides a comprehensive analysis of the aftermath of Hurricane Dorian's impact on Grand Bahama Island, specifically addressing the extensive flooding and saltwater intrusion into aquifers, which significantly affected the island's water supply. Through an exploration of Managed Aquifer Recharge (MAR) and reforestation as potential nature-based solutions, the study conducts a thorough technical assessment of MAR, identifying plausible implementation sites. Additionally, it offers insightful financial and cost-benefit analyses, integrating ecosystem services, for both MAR and reforestation strategies.

The study's approach is noteworthy for its emphasis on holistic consideration and sustainability. While not exhaustive, it offers relevant implications for addressing urgent environmental challenges and enhancing the resilience of ecosystems and local communities in Grand Bahama. However, the study would benefit from further clarification and organization of the methodology. Additionally, a more thorough analysis of the results concerning uncertainty is warranted, considering that the findings were derived from limited data.

REPLY: We thank the reviewer for his/her time and the feedback to our proposed manuscript. We have adjusted the organization of the methodology section 2.4 and extended the description after line 155: please see above in our reply to the first reviewer.

Additionally, a more thorough analysis of the results concerning uncertainty is warranted, considering that the findings were derived from limited data.

REPLY: Thank you for raising this point. In response to that, and to points on uncertainty raised above by the first reviewer, we will extend the discussion in the manuscript as follows (after line 394):

To take uncertainty into account in our analyses, we first applied multiple discount rates. In fact, based on past research, discount rate is amongst the most sensitive parameters and is hence an important source of uncertainty (Costanza & Daly, 1992). A low discount rate reduces the devaluation of future effects, favoring policies with long-term benefits and low

present costs, while a high discount rate does the opposite (Dominati et al., 2014; Hanley, 2013). Thus, a low rate values long-term benefits more, whereas a high rate emphasizes short-term benefits (Dominati et al., 2014). This approach allowed us to understand the effects of one of the most relevant uncertainty sources on the results.

However, other sources of uncertainty can be found in our ES assessment and evaluation. First, uncertainty is inherent in all techniques used for ES estimations (Dominati et al., 2014). Costanza et al. (2017) note that imperfect information affects the evaluation of ES, beginning at the process understanding level and extending through the quantification and economic valuation of ES (Dominati et al., 2014). This imperfection stems from limited biophysical and economic data availability (Dominati et al., 2014) or from relying on simplistic assumptions or expert opinion, such as the relationship between land cover, water provision, and land use (Vollmer et al., 2022). Also, the way this imperfection is included in the ES estimations depends on which models and software are chosen. Due to time resources, our analysis used one main software (InVEST) to guide the ES estimations, but others exist. For example, a promising alternative model is the ARtificial Intelligence for Environment & Sustainability approach (ARIES) (Villa et al., 2014). In our study, multiple parameters were chosen deterministically due to data scarcity (e.g., to estimate investments costs or operation costs), and might be adapted in the future to address uncertainty.

Second, the lack of standards for ES modeling, assessment, and valuation, along with the high time and resource demands of sophisticated methods, pose some challenges (Costanza et al., 2017). For example, this can lead to double counting, where provisioning, regulating, or cultural services are counted alongside their supporting services (Costanza et al., 2017). Inappropriate classification often causes this issue (Fisher et al., 2008). In our analysis, we made sure that double counting is not happening.

In terms of challenges related to the inclusion of the ES assessments into an extended CBA, uncertainties include determining when and how to monetize the benefits of a measure (European Centre for River Restoration, 2022), the lack of result validation (Sudmeier-Rieux et al., 2021), and the exclusion of parameters or ES that cannot be expressed in monetary terms (Ruangpan et al., 2020). For example, in our analysis, we were not able to include the spiritual significance of forest areas in Grand Bahama. Potential ways to address these sources of uncertainty in the future are to use the Monte Carlo approach, or simpler methods such as assuming a spatially uniform error or using alternative raster inputs (Hamel & Bryant, 2017; Vining & Weimer, 2010). Additionally, using multiple models to simulate the same process could help assess the effects of conceptual model uncertainty (Hamel & Bryant, 2017).

Additional references:

Costanza, R., & Daly, H. E. (1992). Natural capital and sustainable development. Conservation Biology, 6 (1), 37–46. https://doi.org/10.2307/2385849

Dominati, E. J., Robinson, D. A., Marchant, S. C., Bristow, K. L., & Mackay, A. D. (2014). Natural capital, ecological infrastructure, and ecosystem services in agroecosystems. In

Encyclopedia of agriculture and food systems (pp. 245–264). Elsevier. https://doi.org/10.1016/B978-0-444-52512-3.00243-6

European Centre for River Restoration. (2022). The economics of river restoration. Retrieved June 2, 2022, from https://www.ecrr.org/River-Restoration/Economics

Fisher, B., Turner, K., Zylstra, M., Brouwer, R., de Groot, R., Farber, S., Ferraro, P., Green, R., Hadley, D., Harlow, J., Jefferiss, P., Kirkby, C., Morling, P., Mowatt, S., Naidoo, R., Paavola, J., Strassburg, B., Yu, D., & Balmford, A. (2008). Ecosystem services and economic theory: Integration for policy-relevant research. Ecological Applications, 18 (8), 2050–2067. http://www.jstor.org/stable/27645921

Hamel, P., & Bryant, B. P. (2017). Uncertainty assessment in ecosystem services analyses: Seven challenges and practical responses. Ecosystem Services, 24, 1–15. https://doi.org/10.1016/j.ecoser.2016.12.008

Villa, F., Bagstad, K. J., Voigt, B., Johnson, G. W., Portela, R., Honzák, M., & Batker, D. (2014). A methodology for adaptable and robust ecosystem services assessment. PloS one, 9 (3), e91001. https://doi.org/10.1371/journal.pone.0091001

Vining, A., & Weimer, D. L. (2010). An assessment of important issues concerning the application of benefit-cost analysis to social policy. Journal of Benefit-Cost Analysis, 1 (1), 1–40. https://doi.org/10.2202/2152-2812.1013

Vollmer, D., Burkhard, K., Adem Esmail, B., Guerrero, P., & Nagabhatla, N. (2022). Incorporating ecosystem services into water resources management-tools, policies, promising pathways. Environmental management, 69 (4), 627–635. https://doi.org/10.1007/s00267-022-01640-9

---

## Author Response (AR2)

**26 Jun 2024; Uploaded files validated; by Natascha Töpfer; Notification to the authors:**

Checking your paper, I noticed that your tables contain coloured cells. Please note that this will not be possible in the final revised version of the paper due to HTML conversion of the paper. When revising the final version, you can use footnotes or italic/bold font. For now, the process will continue, but please note that the final version cannot be published by using coloured tables.

REPLY: We have adopted the tables accordingly (colors are removed from the tables).

**Revision, 08 Sep 2024**

**Editor decision: Publish subject to minor revisions (review by editor)**

by Yongping Wei

Public justification (visible to the public if the article is accepted and published):

The manuscript has been improved a lot. Please address the very constructive comments from Reviewer 2, particularly the structure of your introduction section and other detailed comments. In addition, you should address technical corrections.

REPLY: Thank you very much, we have answered to all comments in the following and implemented most of them in the manuscript.

**Report #1, Submitted on 09 Jul 2024, Anonymous referee #3**

For final publication, the manuscript should be:

accepted as is.

Suggestions for revision or reasons for rejection:

I have no concerns about the current version of the manuscript, and recommend be accepted by the HESS journal. It is time to go forward and publish it.

REPLY: We thank the reviewer for the comments and time in reviewing our manuscript.

**Report #2, Submitted on 14 Aug 2024, Anonymous referee #4**

For final publication, the manuscript should be:

reconsidered after major revisions.

Suggestions for revision or reasons for rejection:

General comments:

The paper presents an analysis of measures aimed at restoring groundwater ecosystem services in the Grand Bahamas. A technical assessment of the potential for Managed Aquifer Recharge (MAR) is combined with a cost-benefit analysis (CBA) that applies two different system boundaries (referred to as a financial and an extended CBA). The underlying problem

and the aim of combining a technical assessment and CBA to enable a more holistic analysis to identify sustainable mitigation measures are relevant and fit the journal's scope. An interesting case study is presented, and relevant aspects related to the applied methods and environmental challenges relevant to Grand Bahamas and similar areas are presented. The current structure (e.g., the problem description and aim in relation to how the results are presented) makes it unclear what the manuscript's primary aim and novelty are. For example, according to the title the focus is on groundwater ecosystem service restoration, but since one of the analysed measures is reforestation, which has no (or rather no analysed) effects on the groundwater ecosystem services, it is a bit confusing for the reader. It is of course possible to compare measures with different purposes, but since much of the manuscript focuses on MAR and the drinking water supply, it becomes unclear. Furthermore, it should be more clearly demonstrated how the proposed method enables the extended CBA, i.e. how it overcomes the perceived weaknesses of CBA.

Below, more detailed comments are provided related to the above mentioned aspects and some other details.

REPLY: We thank the reviewer for the constructive feedback and detailed review, we answered the specific comments below. ***Line numbers refer to the revised manuscript, in its version of the tracked changes mode***.

Specific comments:

Abstract: I miss a comment on the generic results and novelty of the paper, i.e. not only the case study results but also the conclusions and contributions related to the methods or similar.

REPLY: We have extended the abstract, accordingly.

Lines 21-25: MAR only provides 10% of the water demand, but reforestation provides 0% even though the extended CBA show that it is profitable. So, it is important to remember what decision criteria are used. Is it to find the most profitable measure that mitigate any of the damage caused by the hurricane or is it specifically to improve the drinking water supply/restore groundwater ecosystem services?

REPLY: we have changed the wording of Line 22 for being more clear (referring the 10% the the MAR scheme mentioned in the sentence before). Additionally, we would like to remind the reviewer that the MAR scheme's main purpose is to respond to the restoration of groundwater ecosystem services, hence the focus on that aspect, while the reforestation measure has the aim of restoring the forest ecosystem. Both measures lead to additional effects.

Line 36: forest, is it not more appropriate to use the term terrestrial ecosystems?

REPLY: we agree and have adjusted the wording for being more general, now reading "Both island inhabitants and terrestrial ecosystems (including forests)…" (line 38).

Line 65: The RO's high energy consumption is highlighted here. Should the social cost of carbon emissions due to energy consumption be included in the CBA, or is the energy production not causing any $CO_2$ emissions?

REPLY: we agree that energy consumption could be a source of $CO_2$ emissions, which could potentially also be included in the CBA. However, including potential $CO_2$ emissions could

be done for other measures as well, and this goes beyond our capacity and the scope of the study. We will consider adding this aspect in extended CBAs in future work.

Line 64-66: You refer to potential future hurricanes and the potential consequences to the existing RO system and other infrastructure. This is used as a motive for additional measures, but to fully consider this a probabilistic and risk-based approach could have been needed where the probability of future hurricanes and the consequences of them could have been included in the CBA (i.e., the consequences differ between the different alternatives depending on if GB is fully dependent on RO or not). I understand this has not been done and cannot be added now, so this is just a comment on how such aspects typically are included in risk-based CBA.

REPLY: Thanks for the comment, we agree that there is a high probability of hurricanes also in the future. As you correctly state, we did not consider a reoccurrence of the hurricane which could lead to, e.g., the destruction of the reforested area, or further salinization of the freshwater lenses. For once because we would not know how destructive the next hurricane would be and second because we wanted to show the costs and benefits of restoring the ecosystem after Hurricane Dorian. With the current results we see that already restoring the ecosystem services are financially not profitable from Hurricane Dorian, and thus we suspect even a more negative assessment for the forest ecosystem. Regarding the groundwater ecosystem, we suggested mitigation and contingency actions, to be designed to withstand another hurricane event, e.g. We agree that a probabilistic considering is highly relevant and interesting. However, we feel that this goes beyond our study; we recommended this task for future work.

Lines 72-74: You state that "NBS are considered cost-effective and viable solutions". Is it not more appropriate to say that they typically are since all types of NBS are not always cost-effective?

REPLY: We agree, we have changed the wording, accordingly.

Lines 75-76 (this also relates to several other parts of the manuscript): It is stated that two NBS measures to mitigate the impacts of Dorina on groundwater ecosystems are analysed. Since a technical feasibility assessment is performed and the focus is on drinking water supply, it is natural to think that the measures aim to improve access to drinking water. However, reforestation has no effect on water supply, so as a reader, I wonder if the aim is to evaluate any measure that may mitigate the impacts of Dorina. Earlier in the manuscript (line 51), you mentioned that measures to mitigate both groundwater and forest ecosystems were implemented. According to the title, the focus is on groundwater. I suggest you clarify the focus of the paper an why the analysed measures are included.

REPLY: We agree with the reviewer, that reforestation measures have not been studied in term of their effect on the freshwater lenses, as stated in line 495-498. Further we agree that we should make a differentiation between the aim of the MAR scheme and the forestation measure. We have rephrased line 77-78, accordingly. In line 516-522 we have already stated that we differentiate between the aim of the two measures. We have considered your input also by rephrasing the paper title.

Lines 81-91: You state that CBA "falls short of adequately monetarize ES". There are numerous examples of how CBA has been applied to consider effects on ES and is it not rather so that the problem is not the method as such but how it is applied and that there may

be a lack of valuation data to properly value economically the ES? You correctly state that all benefits and costs should be included in a CBA. This is, of course, typically not possible, and limitations are done. However, all relevant benefits and costs should be identified, but some may be excluded from the analysis due to different reasons. Based on this, I have two comments: (1) you state that you propose a method combining a technical feasibility assessment with CBA to (as I understand it) overcome the problem of overlooking ES. However, based on the presented results, it is not clear how the feasibility assessment improves the identification of relevant ES; (2) You do not present a thorough identification ES and related costs and benefits. The analysed ES are presented (lines 221-224), but how was this done, did you use CICCES or any other classification system to ensure no ES were overlooked?

REPLY: In line 83-86 we refer to the CBA as it has been conducted often in the past, also based on guidelines and recommendations of what aspects should be included in a CBA. We change wording in line 84 to "the CBA method, and the way it has been recommended to be conducted in the past, falls short to adequately monetarize ecosystems services".

Lines 120-123: Wellfield 6 constitutes approximately 42% of the total abstraction rate (26/11), but you state that it corresponds to 30% of the total demand. Does it have a higher capacity compared to the actual demand? Please clarify this. In the abstract, it is stated that Dorian caused 40% of the island's water supply to become brackish.

REPLY: In the abstract we had rounded the number of water abstraction from wellfield 6 to 40% (we have now added "about", before this percentage). As stated in line 120-121, 11356 $m^3$/day are abstracted from wellfield 6, out of a total average subtraction of 26497 $m^3$/day on GB. This corresponds to 42.86% of total water abstraction from wellfield 6. These 42.86% (rounded up to 43%) of the total water abstraction are supplied to 30% of the water users, as described line 126. Meaning that 30% of the users consume that fraction (43%) of the total abstracted water. We made that clearer, now, by adding the percentage(s) to line 121 and 126.

Line 135: "sustainability measures (e.g., reforestation)", but you only analyze reforestation and no other sustainability measures.

REPLY: We also consider the MAR measures to be sustainability measures and have reformulated this sentence, for being clearer (line 140-141).

Section 2.2. (Figure 2, etc.): The CBA in both part 2 and part 3 shows what is most profitable; the only difference is that different system boundaries have been applied. In the first case, the focus is on the water supply, so e.g. what is most profitable with respect to water supply. Part 3 identifies the measure most profitable from a societal point of view. This does not make part 3 a novel type of CBA, it is a common type of CBA sometimes referred to as social CBA. The manuscript would benefit from a comment on system boundaries, i.e. which parts of society are included in part 2 and how does it differ from part 3 (global perspective or not?). Linked to a previous comment (lines 81-91), I also suggest you elaborate on how/if the feasibility assessment facilitates the CBA in step 3 or not.

REPLY: We already used the phrase "social … CBA" in line 225. We added a comment on system boundaries in line 94-96: "In other words, a standard CBA focusses on the analysis within the boundaries of the technosphere, i.e., the sphere of human-made technologies and systems, while the extended CBA encompasses studies of the measures' effects on hydrosphere, atmosphere, and biosphere, in addition to the technosphere." In our method, the

feasibility assessment is a pre-condition of the CBA. It can facilitate the CBA, as some information on the measures can be collected in advance for the feasibility study.

Section 2.4 and supplementary material (Table S1): Water price is used to value drinking water supply as an ecosystem service. Since there is typically no real market for drinking water, this will most likely not represent the true value of the drinking water supply. Due to lack of useful valuation studies (WTP etc.), this might be a reasonable assumption, but must be discussed since this is a key ES in the case study.

REPLY: We discussed limitations of evaluation methods in section 3.4.2, and especially in line 415-436.

Lines 193-194: You say all costs and benefits were identified, but this is with respect to the applied system boundaries.

REPLY: This was meant in a generic way. We understand this might lead to confusion, so we deleted "all" and added "within the system boundaries" (line 200).

Results section (but also the method): In a CBA, you typically present a reference scenario, and the measures are compared with this (i.e. what costs and benefits the measures cause). The current situation with RO implemented is presented in the manuscript, but RO is included as a measure. Therefore, the reference scenario/alternative is the current system without RO. The manuscript would benefit from a clear presentation of what is used as the reference scenario.

REPLY: Thank you for the comment. We added an explicit comment on the reference scenario in the methodology in line 220-221 and 268-269.

Lines 267: "the demand was calculated", what calculations are done is it not only assumed equal to the abstraction rate for wellfield 6?

REPLY: True: the abstraction rate of Wellfield 6 is assumed to equal the water demand; we have rephrased this part.

Line 268: "30% of the current brackish water supplied" is it not 100% of the supplied brackish water but 30% of the total demand today?

REPLY: We have rephrased it to make it clearer.

Line 285-287: Is the conclusion based on the risk assessment that the final total risk is acceptable or how are the results used?

REPLY: In line 301-302 we wrote: "Based on the prior results… the following criteria for the selection of the most suitable MAR location were defined that also allow risk mitigation: ". With "prior results" we also address the results of the risk assessment; we designed risk mitigation measures in Section S3 (Supporting Information) and defined a residual risk after the mitigation measure. The identified and elaborated criteria that follow in line 302-306 are already including the mitigation measures to decrease the risks. We now have mentioned Section S3 directly there (line 301), for making this clearer.

Lines 314-315: An analysis of the distribution of costs and benefits in society is typically part of a CBA. Hence, this aspect can be considered in a CBA to make sure there is not an unreasonable distribution.

REPLY: We modified the text to: "Additionally, the question who would take over the costs for the RRWH schemes would need to be discussed as part of the CBA." (line 328-329)

Line 325: has the option for recovering costs been considered in the analysis?

REPLY: No, this has not been included in the analysis.

Line 334: Here, the results are presented for a 4% discount rate, but in the supplementary material a discount rate interval from 1 to 10% is presented (and the results are also presented for this interval). 10% is an extremely high discount rate for a project like the one presented here. More recent recommendations from national authorities recommend a lower rate and in the document referred to (Floy, 2013), it is mentioned that discount rates ranging from 1 to 10% are reported in the literature, but the author recommends "2.2% for benefit-cost analysis with low and high values of 1.5% and 3.5% for sensitivity analysis". If keeping the range 1-10%, please comment on what range is commonly applied.

REPLY: We consider 1 to 10% to display a higher range sensitivity analysis. We added the following explanation in line 362-363: "for a set of ten discount rates from 1% to 10%, as a sensitivity analysis, although the value of 10% goes beyond recommended discount rates values"

Lines 343-345: Since no benefits of reforestation are included, it seems to fall outside the scope of the analysis, and as a reader, you wonder why it is included.

REPLY: Water supply benefits are not included for reforestation, but reforestation brings other benefits (carbon sequestration, habitat quality, timber production) shown in the extended CBA. The mentioned sentence is used to highlight the difference between standard and extended CBA.

Lines 350-352: It is quite a strange comment since you earlier stated that it has no benefits, so it will of course not perform well in a CBA with the system boundaries applied.

REPLY: As replied above, this comment was used to highlight the difference between standard and extended CBA for reforestation.

Section 3.2 and 3.3: As earlier commented, MAR and reforestation have different purposes, and I suggest this is included when the results are discussed.

REPLY: We agree with your comment and tried made this more clear in the manuscript (see replies above). We feel that we are describing this in more detail in section 3.5 and 3.6, where we discuss the different purposes and aspects of MAR compared to RO and reforestation.

Line 409-: You only consider uncertainties in the discount rate, but the uncertainties in the model assumptions should be discussed. To what extent can the applied data for valuing the ES affect the results etc.?

REPLY: We extensively discuss limitations of the CBA methods later in section 3.4.2.

Lines 434-435: How do you ensure there is no double-counting?

REPLY: We could do that by using the MEA (2005) classification. We added this comment to line 448.

Lines 468-470: If this is included, the results may look different, so it is an important uncertainty to comment on when discussing model uncertainties.

REPLY: Thank you, we added the comment on potential additional uncertainties related to this aspect in line 483.

Line 506: You write two measures, but you include and analyse three measures since RO is considered a measure in the analysis.

REPLY: We feel that "the third one", RO, is taken up two sentences later. We chose this wording in order to highlight the aspect of sustainability (which we see for MAR and reforestation, but not to RO).